# Structural and functional dissection of the DH and PH domains of oncogenic Bcr-Abl tyrosine kinase

Sina Reckel[1], Charlotte Gehin [2], Delphine Tardivon[1], Sandrine Georgeon[1], Tim Kükenshöner[1], Frank Löhr[3], Akiko Koide[4,5,6], Lena Buchner[3], Alejandro Panjkovich[7], Aline Reynaud[8], Sara Pinho[1], Barbara Gerig[1], Dmitri Svergun[7], Florence Pojer[8], Peter Güntert[3,9,10], Volker Dötsch[3], Shohei Koide [4,5,6], Anne-Claude Gavin[2] & Oliver Hantschel [1]

The two isoforms of the Bcr-Abl tyrosine kinase, p210 and p190, are associated with different leukemias and have a dramatically different signaling network, despite similar kinase activity. To provide a molecular rationale for these observations, we study the Dbl-homology (DH) and Pleckstrin-homology (PH) domains of Bcr-Abl p210, which constitute the only structural differences to p190. Here we report high-resolution structures of the DH and PH domains and characterize conformations of the DH–PH unit in solution. Our structural and functional analyses show no evidence that the DH domain acts as a guanine nucleotide exchange factor, whereas the PH domain binds to various phosphatidylinositol-phosphates. PH-domain mutants alter subcellular localization and result in decreased interactions with p210-selective interaction partners. Hence, the PH domain, but not the DH domain, plays an important role in the formation of the differential p210 and p190 Bcr-Abl signaling networks.

[1] Swiss Institute for Experimental Cancer Research (ISREC), School of Life Sciences, École polytechnique fédérale de Lausanne (EPFL), 1015 Lausanne, Switzerland. [2] Structural and Computational Biology Unit, European Molecular Biology Laboratory (EMBL), 69117 Heidelberg, Germany. [3] Institute of Biophysical Chemistry, Goethe University Frankfurt, 60438 Frankfurt, Germany. [4] Laura and Isaac Perlmutter Cancer Center, New York University Langone Medical Center, New York, NY 10016, USA. [5] Department of Medicine, New York University School of Medicine, New York, NY 10016, USA. [6] Department of Biochemistry and Molecular Pharmacology, New York University School of Medicine, New York, NY 10016, USA. [7] European Molecular Biology Laboratory (EMBL), Hamburg Outstation, 22607 Hamburg, Germany. [8] Protein Crystallography Core Facility, School of Life Sciences, École polytechnique fédérale de Lausanne (EPFL), 1015 Lausanne, Switzerland. [9] Laboratory of Physical Chemistry, ETH Zürich, 8093 Zürich, Switzerland. [10] Graduate School of Science, Tokyo Metropolitan University, Tokyo 192-0397, Japan. Correspondence and requests for materials should be addressed to O.H. (email: oliver.hantschel@epfl.ch)

The Bcr-Abl oncoprotein is expressed from the Philadelphia (Ph) chromosome, which is formed upon the t(9;22) reciprocal chromosomal translocation that fuses the breakpoint cluster region (BCR) gene with the Abelson tyrosine kinase (ABL1)[1]. Bcr-Abl is a constitutively active tyrosine kinase and is selectively inhibited by imatinib (Gleevec), which became a paradigm for targeted cancer therapy[2]. Different Bcr-Abl protein isoforms are expressed, all of which contain exons 2–11 of the ABL1 gene, but, depending on the location of the translocation breakpoint in BCR, include different portions of the BCR gene[3]. The most common Bcr-Abl isoforms are p210 and p190. p190 lacks residues 427–927 of Bcr, and thus is 501 amino acids (i.e., ~25%) shorter than p210, but otherwise contains the same domains with identical sequences (Fig. 1)[4]. Bcr-Abl p210 is the molecular hallmark of chronic myelogenous leukemia (CML)[3]. In addition, 20–30% of adult B-cell acute lymphoblastic leukemias (B-ALL) are Bcr-Abl-positive, of which approximately 1/4 of cases express p210, whereas 3/4 express Bcr-Abl p190. The treatment of CML patients with the Bcr-Abl tyrosine kinase inhibitor (TKI) imatinib results in deep remissions and long-term survival[5]. Although TKIs prolong survival in Bcr-Abl-positive B-ALL, acquired TKI resistance is frequent and overall survival is still dramatically low[6, 7]. Therefore, a deeper understanding of Bcr-Abl p210 and p190 signaling may guide the identification of additional drug targets for combinatorial therapy in B-ALL.

We have recently performed a quantitative comparative proteomics study of Bcr-Abl p210 and p190 in parallel to a second laboratory[8, 9]. Unexpectedly large differences in the interactome and tyrosine phosphoproteome were found, with more than a dozen proteins that preferentially interact with either p210 or p190 and ~100 differential phosphotyrosine (pY) sites indicating activation of distinct signaling pathways by the two Bcr-Abl isoforms[8, 9]. In contrast, no difference in degree and sites of p210 and p190 autophosphorylation or in vitro kinase activity of Bcr-Abl was detected that could have rationalized the observed signaling differences. These observations suggested that the functional disparities of p210 and p190 arise from regions outside the kinase domain. Hence we decided to study the differences in structure of the two isoforms. Due to the different translocation breakpoints, p210 contains a predicted Dbl-homology (DH) and Pleckstrin-homology (PH) tandem domain that p190 lacks (Fig. 1). These two domains constitute the only structurally uncharacterized domains of Bcr-Abl and their functions have not been characterized in detail.

DH–PH tandem domains are the canonical structural motifs of the Dbl-family guanine nucleotide exchange factors for Rho GTPases (RhoGEF), with the DH domain mediating GEF activity[10]. Previous work has indicated differential activation of Rho GTPases RhoA, Rac1 and Cdc42 in Bcr-Abl p210-, as well as in p190-expressing cells[11, 12]. In this context, Rac1 and Cdc42 activation was proposed to be dependent on the RhoGEF Vav,

which is activated downstream of both p210 and p190[12]. It has also been shown that a point mutation in the predicted Bcr-Abl DH domain (S509A) that was proposed to lack GEF activity was able to induce leukemia with shorter latency as compared to wild-type p210 in a mouse model[13]. Thus far, the activity and selectivity of the Bcr-Abl DH domain remains unclear and no convincing direct link between Rho GTPase activation and the Bcr-Abl DH domain could be made.

PH domains are well-characterized lipid-binding domains found in a variety of proteins, in which they mediate interactions with phosphoinositide lipids in biological membranes[14]. As part of the DH–PH unit, the PH domain has been implicated in multiple functions, including membrane localization, allosteric modulation of GEF activity, and in certain cases direct contributions to DH GEF activity[15]. Deletion of the entire PH domain from Bcr-Abl p210 slightly increased transformation and mildly decreased survival in mouse bone marrow transplantation models, showing a disease phenotype that resembles p190-driven B-ALL[16].

In contrast to the DH and PH domains, the other domains of Bcr-Abl have been structurally well characterized, including the central SH3-SH2-kinase domain unit in both the autoinhibited form[17–19] (reviewed in ref. [20], Fig. 1) and the active form[19, 21–24] revealing critical and targetable mechanisms for allosteric regulation of the kinase (reviewed in ref. [25]). Crystallographic analysis of the N-terminus of Bcr-Abl showed that a coiled-coil domain forms antiparallel homo-tetramers[26] and the very C-terminus of Bcr-Abl contains an F-actin binding domain that has been studied in solution by NMR[27].

In summary, the absence of a structural model and poorly defined biochemical specificity and regulation of the Bcr-Abl DH and PH domains, in combination with conflicting data on the role of both domains in Bcr-Abl-dependent leukemogenesis and signaling necessitates an in-depth structural and functional characterization. Here we provide high-resolution structures of the DH and PH domains that unravel unique structural features. Detailed functional characterization give no evidence for RhoGEF activity of the DH domain. In contrast, we characterize in detail PH-mediated phosphoinositide binding that affects cellular localization and the signaling network of Bcr-Abl p210.

## Results

**Structure of DH domain by NMR and X-ray crystallography.** The DH and PH domains constitute the only predicted folded domains of Bcr-Abl whose structure remains undetermined. Therefore, we set out to structurally characterize them by crystallography. Among several constructs, the Bcr-Abl DH- (residues 487–702) and DH–PH-domain (residues 487–893) constructs expressed in high yields and showed good biophysical behavior (Supplementary Fig. 1). However, we were unable to obtain

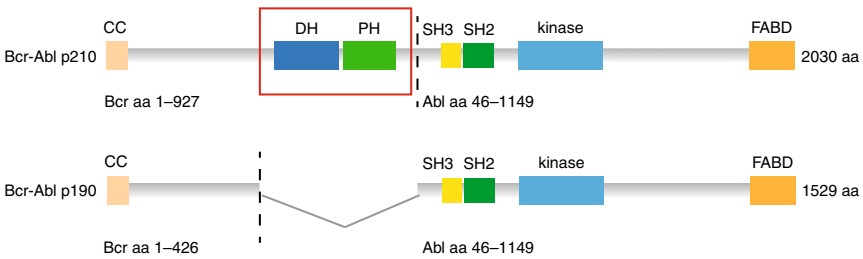

**Fig. 1** Domain organization of Bcr-Abl. The two isoforms of the fusion protein Bcr-Abl, p210, and p190, are shown with their sizes and domain arrangement. The breakpoint between Bcr and Abl is indicated with a dotted line. CC coiled-coil, DH Dbl-homology, PH Pleckstrin-homology, SH3/SH2 Src-homology 3/2, FABD F-actin binding domain

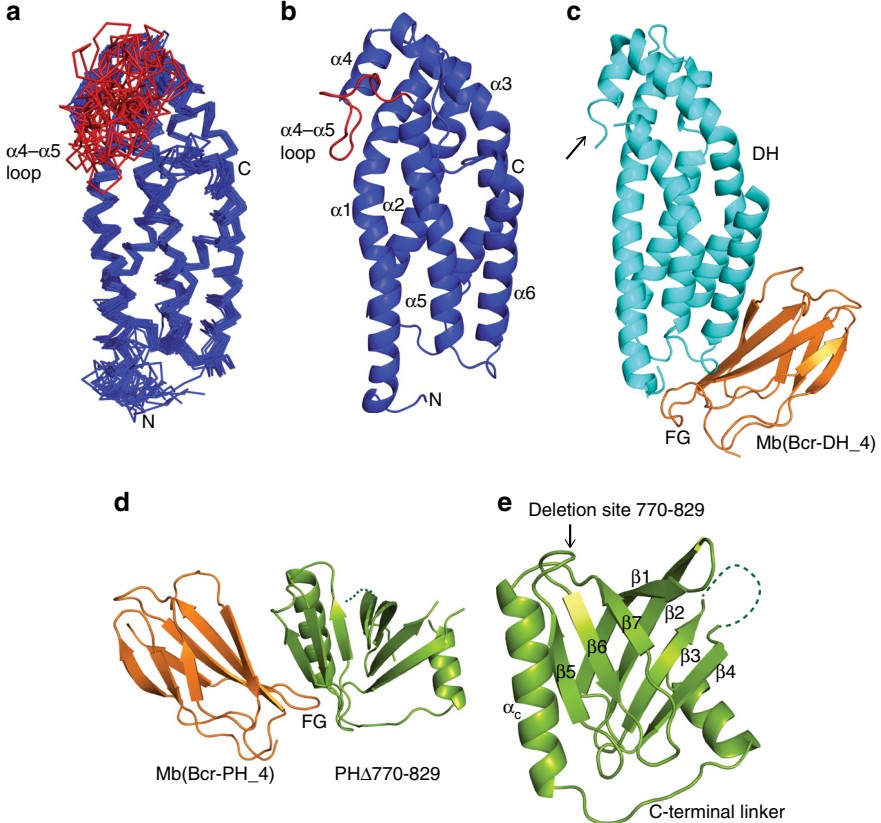

**Fig. 2** Structures of the DH and PH domains by NMR and X-ray crystallography. **a** Bundle of 20 DH structures as determined by NMR spectroscopy. The long and flexible α4–α5 loop region (residues 622–638) is highlighted in red. **b** Cartoon representation of the NMR conformer closest to the mean. The six helical elements and the α4–α5 loop region are labeled. **c** Crystal structure of the DH domain in complex with Mb(Bcr-DH_4) in cartoon representation. The α4–α5 loop region is missing in this structure (black arrow). The monobody interacts with the DH domain via its FG loop, which is labeled. **d** Crystal structure of the internally deleted PH domain (PHΔ770–829) in complex with Mb(Bcr-PH_4) in cartoon representation. The monobody interacts with the PH domain via its FG loop, which is labeled. **e** Cartoon representation of the PH domain (PHΔ770–829) with assignment of the secondary structure elements, the internal deletion site covering residues 770–829 and the C-terminal linker region

crystals for any of our constructs, so we turned to nuclear magnetic resonance (NMR) spectroscopy in order to get a better idea about the folding and flexibility of these proteins. The [$^{15}$N,$^1$H]-TROSY spectrum of the DH domain was clearly resolved and was indicative of a well-folded protein. We therefore embarked on the structure determination of this domain by NMR. Given the considerable size of the DH–PH tandem domain (410 residues, 47 kDa) and poor behavior of the isolated PH domain, we first concentrated on the DH domain (219 residues, 25 kDa).

The Bcr-Abl DH domain NMR structure was based on 4298 NOE upper distance limits, of which 845 were long-range (Supplementary Table 1, BMRB entry 34101). The 20 structures with the lowest target function nicely overlay with an average backbone atom root-mean-square deviation (RMSD) of 0.85 Å in the folded regions (PDB ID 5N6R, Fig. 2a). The DH domain forms a six-helix bundle arranged with a kink toward one side similar to the 'seat-back of a chaise longue' in line with the canonical fold of DH domains (Fig. 2b, Supplementary Table 1, Supplementary Fig. 2)[28]. An interesting feature that is specific to the Bcr-Abl DH domain is the extended 18-residue-long loop region between α-helices 4 and 5 covering residues 622–638, which is 10–13 residues longer than in most other DH domains. The α4–α5 loop of the Bcr-Abl DH domain is very dynamic and does not adopt a preferred conformation (Fig. 2a). Apart from this loop, only the N-terminal and C-termini of the protein are disordered.

To increase the crystallization probability of the DH–PH tandem domain, we generated monobodies, synthetic binding proteins based on the fibronectin type III domain that bind to the DH–PH domain[29, 30] (Supplementary Fig. 1). Monobodies tightly bind to the target protein in a conformation-specific manner and are often effective crystallization chaperones that stabilize a low-energy state among the native conformational ensemble and create productive crystal contacts[31, 32]. Initial screening of monobodies generated using the DH–PH tandem domain as the target yielded only clones (termed Mb(Bcr-DH_1–4)) that bound to the DH domain with low nanomolar dissociation constants ($K_d$, Supplementary Fig. 3). Crystallization trials using these monobodies in complex with the DH domain readily gave crystals for the DH/Mb(Bcr-DH_4) complex that diffracted to 1.65 Å and enabled the structure determination (PDB ID 5N7E, Supplementary Table 2).

The crystal structure of the DH domain includes the six major α-helices in accordance with the NMR structure (Fig. 2). No electron density was observed for the extended flexible α4–α5 loop region seen in the NMR structure, so this loop was not included in the crystallographic model. Overall the comparison of the DH NMR- and crystal-structures revealed an identical overall fold with some variation in helix orientations (Supplementary Fig. 2). The monobody bound, as expected, via diversified positions in the FG loop and the βC/βD strands, to the tip of the DH domain proximal to the N-terminus (Fig. 2c, Supplementary Figs. 2,3). To investigate the impact of monobody binding on the DH domain structure, we compared the NMR chemical shift changes upon formation of a complex between the unlabeled

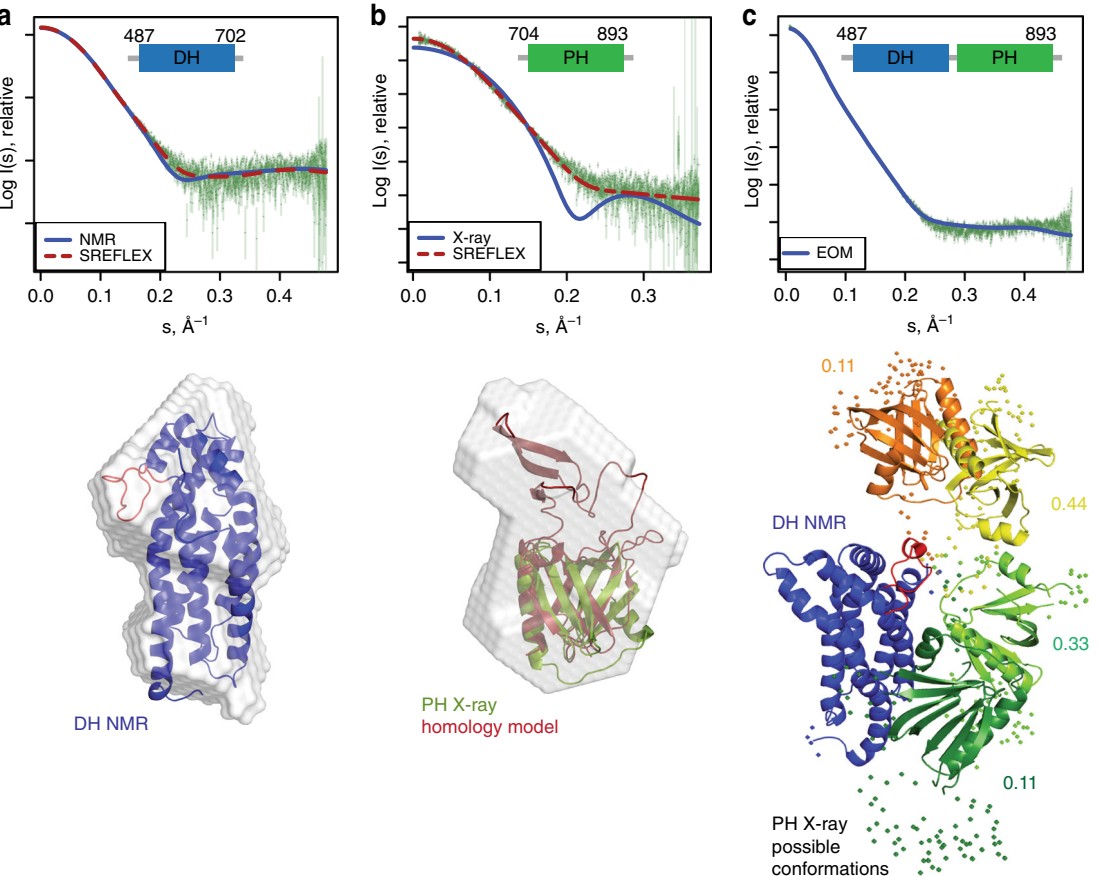

**Fig. 3** Small-angle X-ray scattering (SAXS) of the DH–PH tandem domain. **a** Experimental SAXS profile of the DH domain (green dots) and theoretical scattering profiles for the NMR model (discrepancy $\chi^2 = 1.5$) and SREFLEX refined model (discrepancy $\chi^2 = 1.0$). Below, the DH domain NMR structure is superimposed with the SAXS ab initio model as gray envelope. The $\alpha4$–$\alpha5$ loop is highlighted in red. **b** Experimental SAXS profile of the PH domain (green dots) and theoretical scattering profiles of PH crystal structure (blue, discrepancy $\chi^2 = 8.1$), as well as a refined model for the full-length PH domain (red, discrepancy $\chi^2 = 1.1$). Modeling of the full-length PH domain was based on a homology model with Xpln as a template (PDB ID 2Z0Q). For the PH domain, the SAXS based ab initio model is shown superimposed with the refined model for the full-length PH domain (red) and the PH$\Delta$770–829 crystal structure (green). **c** Experimental SAXS profile of the DH–PH domain (green dots) and theoretical scattering of a representative solution ensemble after EOM (blue, discrepancy $\chi^2 = 1.0$). Possible conformations of the DH–PH tandem domain are shown after superposition on the DH domain (blue). The respective orientations of the PH domain are represented in orange, yellow, light, and dark green, and the fractions for the respective PH domain positions are indicated next to the structures illustrating the inter-domain flexibility. Regions without high-resolution structural data, including the DH–PH linker and the internal PH deletion 770–829, are shown as spheres

monobody and the [15]N-labeled DH domain. We observed large chemical shift changes throughout the N-terminal half of the DH domain, whereas the most distal region relative to the monobody binding epitope remained largely unperturbed (Supplementary Fig. 4). In summary, the solution and crystal structures of the Bcr-Abl DH domain showed the canonical six helix-fold, but with a long $\alpha4$–$\alpha5$ loop.

**PH domain has canonical fold with large insertion.** The Bcr-Abl PH domain exhibited limited stability at room temperature despite extensive optimization efforts. Homology modeling indicated the presence of a unique insertion of 59 amino acids (residues 770–829) between predicted strands $\beta5$ and $\beta6$ constituting almost a third of the entire Bcr-Abl PH domain. Since the presence of such a large, putatively flexible element can dramatically interfere with any structure determination approaches, we deleted this part of the PH domain. Still, we were not able to obtain crystals. Encouraged by our success with the DH domain, we generated additional monobodies that bound to the PH domain. Here we made use of the phage-display pool of

monobodies originally enriched with the Bcr-Abl DH–PH tandem domain, but performed the secondary yeast-display screening using the PH domain as the target (Supplementary Fig. 1). We obtained three different monobodies with low nanomolar $K_d$ values (termed Mb(Bcr-PH_2–4), Supplementary Fig. 3). These monobodies bound both the full-length and the shortened PH domain (PH$\Delta$770–829) with equal efficiency (Supplementary Fig. 3). With these new constructs and tools in hand, we obtained crystals for the full-length PH domain in complex with monobody Mb(Bcr-PH_4), but we could not optimize these crystals beyond a resolution of 8 Å. In contrast, crystals of PH$\Delta$770–829 in complex with the same monobody diffracted up to 1.65 Å and allowed structure determination of this complex (Fig. 2d).

The structure of the PH domain comprises the canonical PH domain fold with a $\beta$-sandwich, in which four $\beta$-strands at the N-terminal part of the protein pack against a 3-stranded C-terminal $\beta$-sheet, and a long C-terminal $\alpha$-helix (Fig. 2e), displaying high structural similarity for example to the PH domain of Sos1 (Supplementary Fig. 2). In addition, our construct further includes a C-terminal helical linker that was required for the stability of the PH domain. This linker leads toward the fusion

breakpoint with the Abl kinase and connects it to the remaining Cap region and SH3 domain in Abl[20], whereas in the full-length Bcr protein this linker would connect the PH domain to a C2 domain. Based on our biochemical data, the PH domain forms a 1:1 complex with the monobody, which binds to the β4–β5 loop and the αC helix of the PH domain via its variable FG loop (Supplementary Fig. 3).

**SAXS analysis of DH and PH domains.** To obtain a better understanding of the interplay of the DH and PH domains, we used small-angle X-ray scattering (SAXS). In other DH–PH units, the PH domain may regulate GEF activity either via intramolecular interactions with the DH domain, which could block GTPase binding or by forming an additional interface with the GTPases to facilitate nucleotide exchange[28, 33].

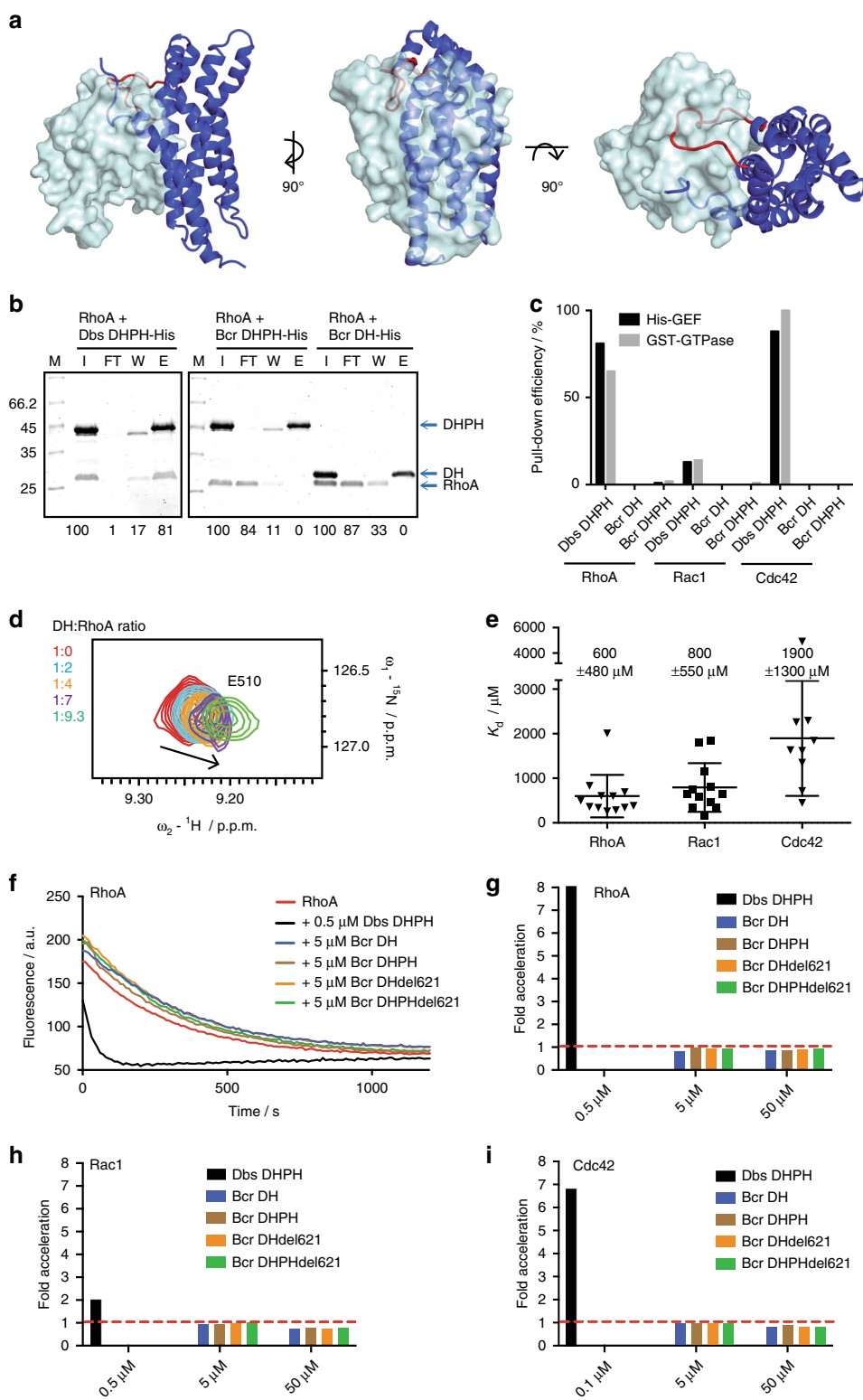

SAXS data for the DH, PH and DH–PH domains allowed a detailed analysis of the conformations of the constructs in solution (Supplementary Table 3, Supplementary Fig. 5). As a control, SAXS data of the isolated DH domain agreed well with the NMR and X-ray structure (Fig. 3a). Since the SAXS data of the PH domain were obtained for the full-length PH construct, fitting with the PH crystal structure with the deleted β5–β6-insertion was unsatisfactory (Fig. 3b). We therefore made a homology model for the full-length PH domain and fitted it with the SAXS data using SREFLEX[34]. The resulting model indicated that residues 770–829 of the Bcr-Abl PH domain protrude away from the core of the globular PH domain and add a significant volume to the overall shape of the domain (Fig. 3b). The insertion therefore does not seem to affect the PH fold nor does it interact extensively with the core PH domain. It could possibly provide an additional surface for protein–protein interactions.

The DH–PH tandem domain data were fitted using the NMR structure of the DH domain and the crystal structure of the PH domain and further analyzed using the ensemble optimization approach (EOM)[35] to evaluate the inter-domain flexibility of the two domains (Fig. 3c, Supplementary Table 3, Supplementary Fig. 5). The scattering profile of the DH–PH tandem domain in solution fitted best to an ensemble of four different conformations with a flexibility of 75% ($R_{flex} = 0.75$, discrepancy $\chi^2 = 1.0$, Fig. 3c), where 100% corresponds to maximum flexibility. This high flexibility between the DH and PH domains argues for a balls-on-a-string model with rather independent behavior without specific domain–domain interaction interfaces. These findings are also in agreement with results from an NMR experiment, in which only minimal chemical shift perturbations of $^{15}$N-labeled DH domain were observed upon addition of unlabeled PH domain in *trans* (Supplementary Fig. 4). We therefore conclude that the DH and PH domains function as independent units and have only minimal influence on each other.

**The DH–PH domain has no detectable RhoGEF activity**. For classical Rho GEF family members, the interaction with GTPases is mediated by two conserved regions, CR1 and CR3, which are part of the helical core of the DH domain, as well as the C-terminal α-helix 6 (Supplementary Fig. 2)[36]. In particular a glutamate residue in CR1 and a lysine residue in CR3 are important for catalytic activity[36–38]. Furthermore, residues in the 'seat-back-region' of the DH domains, comprising α-helix 4, α4–α5 loop and α-helix 5, determine GTPase specificity[15]. Notably, these residue that are important for GTPase interaction, catalysis, and selectivity are not conserved in the Bcr-Abl DH domain. Furthermore, the extended α4–α5 loop points directly into the GTPase binding interface, where it would sterically clash with binding of a Rho GTPase (Fig. 4a, Supplementary Fig. 2).

To probe our structural predictions, we first measured the interaction of the Bcr-Abl DH–PH tandem domain with the three canonical Rho GTPases RhoA, Rac1, and Cdc42. To account for a possible regulatory effect of the PH domain, all experiments were conducted with the DH domain alone, as well as the DH–PH tandem. As a positive control, we included the well-characterized RhoA- and Cdc42-selective DH–PH domain of Dbs[15].

Since a number of DH–PH domains, such as Dbs, Intersectin, Tiam1, Trio, and LARG have been shown to form stable complexes with GTPases and have been co-crystallized, we first tested the interaction with recombinant RhoA, Rac1, and Cdc42 using in vitro pull-down experiments and co-migration on a size-exclusion column (Fig. 4b, c, Supplementary Fig. 6). As expected, the Dbs DH–PH domain showed very efficient pull-down of and co-elution with its cognate GTPases RhoA and Cdc42, and to a lesser extent Rac1 (Fig. 4b, c, Supplementary Fig. 6). In contrast, the Bcr-Abl DH or DH–PH domain constructs did not display detectable interactions with any of the three Rho GTPases (Fig. 4b, c, Supplementary Fig. 6). We concluded that there is no high-affinity interaction between the Bcr-Abl DH domain and the tested Rho GTPases. These observations are in line with our previous proteomics study, in which we did not detect Rho GTPases as interactors with Bcr-Abl p210[8]. We also conducted tandem-affinity purifications (TAP) mass spectrometry experiments in the Bcr-Abl p210-expressing cell line K562 using the isolated DH–PH tandem domain to assess direct interactors of this fragment, but we found no Rho GTPase captured with the Bcr DH–PH tandem domain (Supplementary Data 1).

To exert the function of a GEF, however, a weaker or more transient interaction of the DH–PH domain with Rho GTPases may suffice. To explore this possibility, we conducted NMR titration experiments, in which we titrated RhoA, Rac1 or Cdc42 to the $^{15}$N-labelled DH domain (Fig. 4d, e, Supplementary Fig. 7). Overall, chemical shift perturbations were rather small and visible only upon addition of excess GTPase, excluding large conformational changes in the DH domain upon interaction with the respective GTPase. Additionally, line broadening at higher GTPase concentrations challenged the analysis. Nonetheless, about 10 peaks for each experiment showed concentration-dependent chemical shift perturbations throughout the titration series, which were employed for $K_d$ determination. We obtained mean $K_d$ values of 600 μM for the interaction with RhoA, 800 μM for Rac1 and 1900 μM for Cdc42 (Fig. 4e). We furthermore mapped the chemical shift perturbations on the DH domain structure (Supplementary Fig. 7). Interestingly, all three GTPases induced a similar pattern of chemical shift perturbation with an accumulation around the putative GTPase binding site. Collectively, these experiments indicated that the interactions with the three Rho GTPases are specific, but of much lower affinity than other GTPase-GEF interactions, which are typically in the low-micromolar range. Although the physiological significance of

**Fig. 4** Functional characterization of the DH domain. **a** Model for a complex of RhoA with the DH domain of Bcr-Abl. The GTPase binding model was inferred from the co-crystal structure of Dbs and RhoA (PDB ID 1LB1) upon alignment of the Bcr-Abl DH domain with the Dbs DH domain. The α4–α5 loop in Bcr-Abl sterically clashes with RhoA in this model. **b** Representative SDS-PAGE analysis of the Ni-affinity pull-down experiment using His-tagged DH–PH or DH constructs and RhoA. The prey pull-down efficiency is calculated from signal quantification normalized to the input and shown below the respective lane. I input, FT flow-through, W wash, E elution. **c** Summary of the pull-down efficiency using either His-tagged DH–PH constructs or GST-tagged GTPases. **d** Chemical shift perturbation of E510 backbone NH during the NMR titration of the $^{15}$N-labeled DH domain with increasing amounts of RhoA. **e** Binding affinities for interaction of the three GTPases RhoA, Rac1 and Cdc42 with the DH domain derived from fitting the chemical shift changes with increasing GTPase concentration (Supplementary Fig. 6). Mean $K_d$ values ± s.d. are indicated. **f** Nucleotide exchange experiment for mant-GDP-loaded RhoA. The decrease of the fluorescence was monitored in the presence of different Bcr-Abl DH–PH constructs and the Dbs DH–PH as a positive control. Experiments were done in duplicates at two different concentrations of the Bcr-Abl DH and DH–PH constructs. **g** Acceleration of the RhoA nucleotide exchange, i.e., the rate constant in the presence of a DH–PH construct compared to the RhoA intrinsic nucleotide exchange rate, plotted for the different DH–PH constructs at the respective concentrations. **h, i** Acceleration of Rac1 **h** and Cdc42 **i** nucleotide exchange for the different DH–PH constructs at the respective concentrations. Raw data are shown in Supplementary Fig. 8

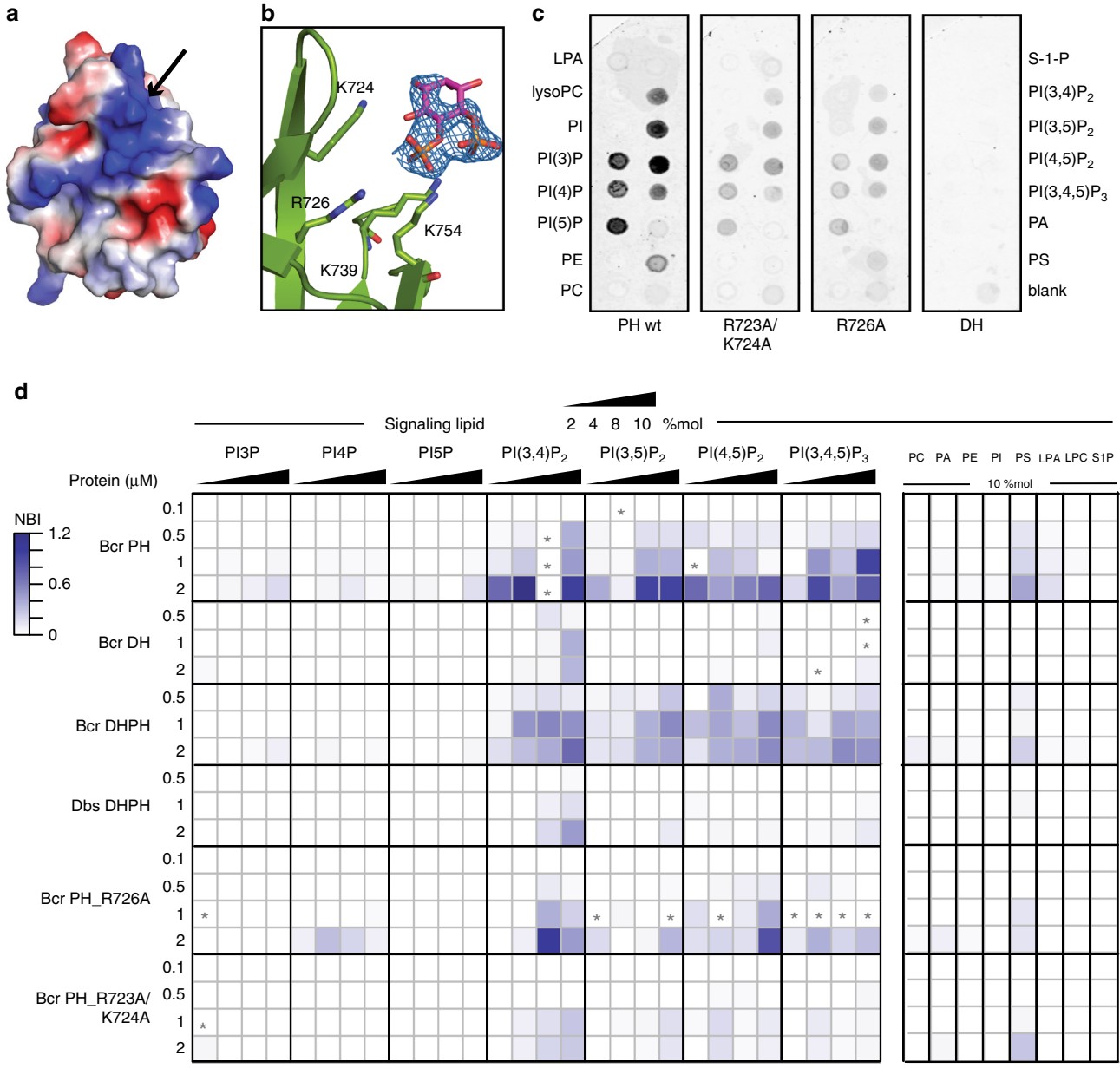

**Fig. 5** Characterization of PIP binding to the PH domain. **a** Structure-based identification of the PIP-binding motif. The electrostatic surface representation indicates that the PIP-binding motif (black arrow) forms a positively charged patch on the PH surface. **b** The PIP-binding site is shown enlarged together with a D-myo-inositiol-4,5-bisphosphate (IP₂) ligand fitted in the 2Fo–Fc electron-density map, contoured at 1.0 sigma. Critical residues in the binding motif are shown in sticks representation and colored by element. **c** PIP strips comparing the PIP-binding properties of PH wt, the PH R723A/K724A and R726A mutants, as well as the DH domain as a negative control. Individual signals on the strip were quantified relative to PH wild-type protein. **d** Binding of Bcr-Abl domains to surrogates of biological membranes in the liposome microarray (LiMA) assay. 0.1, 0.5, 1, and 2 μM of purified sfGFP-tagged Bcr-Abl DH, PH, DH–PH or Dbs DH–PH domains were incubated together with giant unilamellar liposomes containing the indicated concentrations of signaling lipid (in mol % of total lipids). PI3P, PI4P, PI5P, PI(3,4)P₂, PI(3,5)P₂, PI(4,5)P₂, and PI(3,4,5)P₃ are phosphoinositides (Supplementary Data 2, 3). PC phosphatidylcholine, PA phosphatidic acid, PE phosphatidylethanolamine, PI phosphatidylinositol PS phosphatidylserine, LPA lyso PA, LPC lyso PC, S1P sphingosine-1-phosphate, NBI normalized binding intensity. Not determined values are indicated with a star (*). Values are means (n = 3)

these weak interactions is not clear, the observations encouraged us to probe the catalytic GEF function of the Bcr-Abl DH–PH domain in nucleotide exchange experiments. Using fluorophore-labeled GDP, the GDP to GTP exchange was monitored in the presence of the Bcr-Abl DH or DH–PH domains, the Dbs DH–PH fragment as the positive control, and we further included a Bcr-Abl DH loop-deletion mutant into our experiments, in which we shortened the long unstructured α4–α5 loop by 8 residues (DHΔ621–628) to assess a possible steric hindrance of this long loop region (Fig. 4a, f, Supplementary Fig. 8). While the

Dbs DH–PH lead to a strong acceleration of nucleotide exchange of RhoA and Cdc42 and to lesser extent for Rac1 at concentrations as low as 0.1 μM, none of the Bcr-Abl DH, DH–PH and DHΔ621–628 constructs used in these experiments showed an enhancement of the nucleotide exchange rate for any of the three GTPases, even at 10–100-fold higher concentration than in the experiment with the Dbs control (Fig. 4f–h, Supplementary Fig. 8).

Collectively, our results show that even though the Bcr-Abl DH–PH fragment interacts with very low affinity with the three

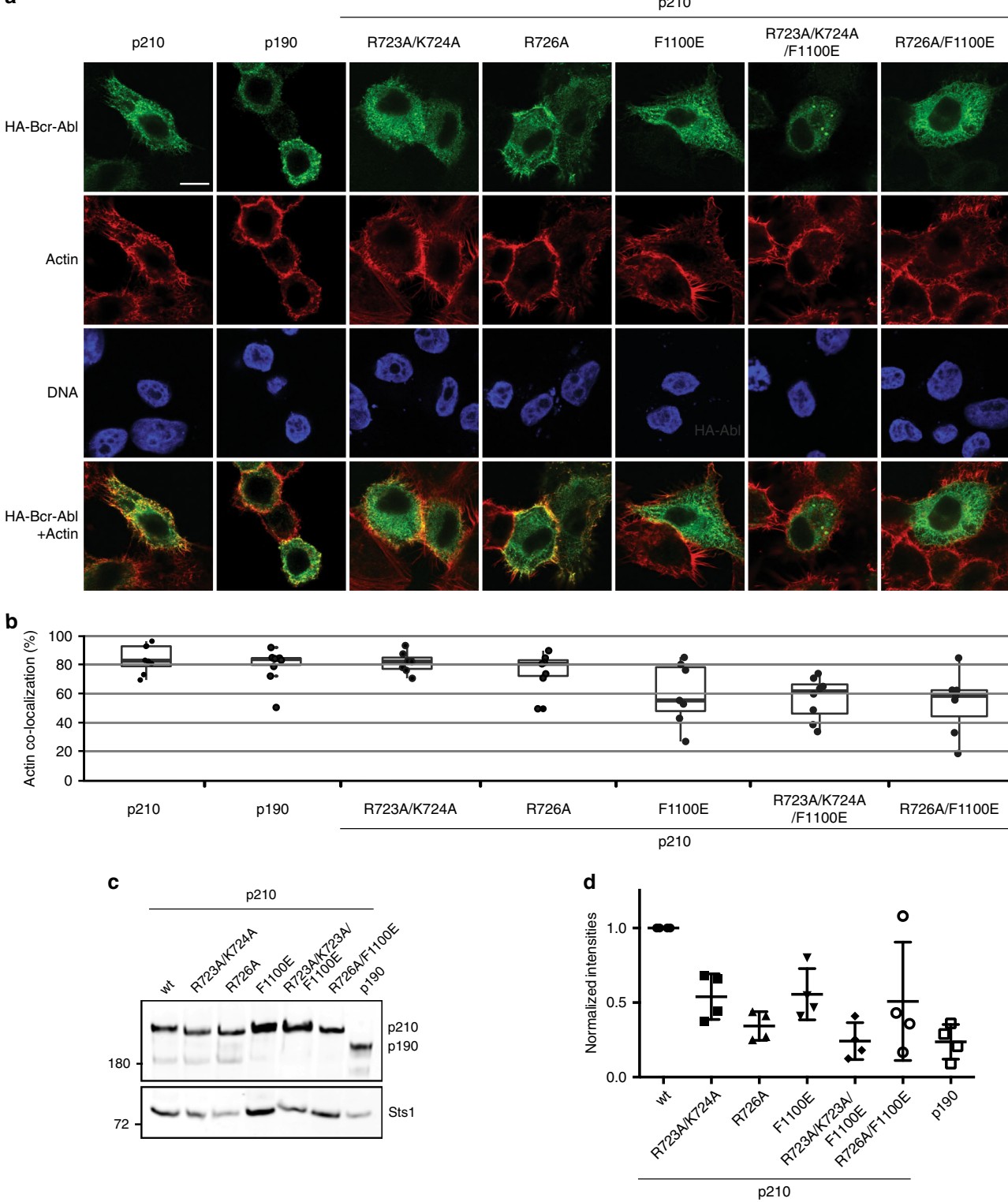

**Fig. 6** Localization and signaling network of Bcr-Abl p210, p190, FABD and PH mutant proteins. **a** Cellular localization of Bcr-Abl p210 and p190, p210 with the PH mutations R723A/K724A or R726A, with the FABD mutant F1100E, as well as a combination of PH and FABD mutation. Cells were fixed and immunostained with an anti-HA antibody (HA-Bcr-Abl). F-actin (actin) and nuclei (DNA) were stained with rhodamine-conjugated phalloidin and Hoechst, respectively. The white scale bar in the upper left image corresponds to 10 μm. **b** Quantification of co-localization of Bcr-Abl with actin fibers using ImageJ. The box-and-whisker plots show the median, lower/upper quartile, and minimum/maximum of % Actin co-localization for at least six quantified cells per Bcr-Abl mutant. **c** Representative immunoblot analysis of anti-Bcr-Abl/Abl immunoprecipitates from BaF3 cells stably transduced with the indicated Bcr-Abl variants. Fractions of the immunoprecipitates were immunoblotted for Bcr-Abl and its interaction partner Sts1. **d** Quantification of the co-immunoprecipitation of Sts1 with Bcr-Abl. The quantified amount of co-immunoprecipitated Sts1 after correction for the Bcr-Abl amounts is shown. Means ± s.d. of two biological replicates analyzed in two technical replicates

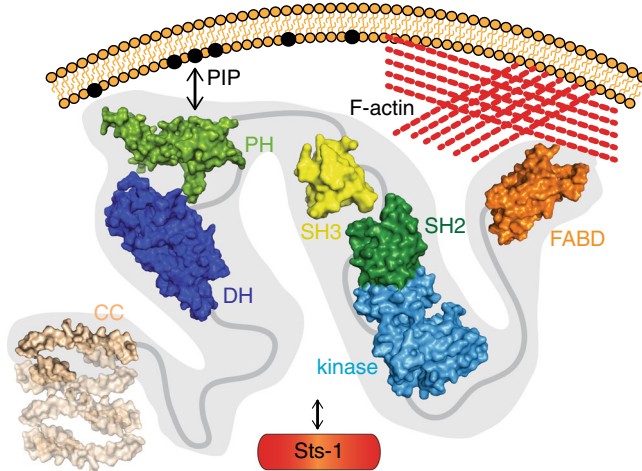

**Fig. 7** Model of the structural organization of Bcr-Abl p210 and contributions to cellular localization and signaling. Domain CC coiled-coil, DH Dbl-homology, PH Pleckstrin-homology, SH3/SH2 Src-homology 3/2, FABD F-actin binding domain. High-resolution structures of the individual domains were solved in this study (DH: PDB ID 5NR6, PH: PDB ID 5OC7) and in previous studies (CC: PDB ID 1K1F, SH3: PDB ID 1OPK, SH2-Kinase: PDB ID 1OPL chain B, FABD: 1ZZP)

canonical Rho GTPases RhoA, Rac1 and Cdc42, it does not show nucleotide exchange activity, confirming the structural interpretation that this DH–PH domain is not a classical RhoGEF family member, but rather resembles a pseudo-GEF.

**PH domain binds phosphoinositides with high affinity.** To test for PH domain function, we first aimed to define the phosphatidylinositol-phosphate (PIP)-binding site, which is formed by a positively charged patch of amino acids (arginine and lysine) in the β2–β3 hairpin of PH domains[39]. In the Bcr-Abl PH structure residues K724 and R726 in the β1–β2 hairpin, in addition to K739 and K754 in the β3–β4 hairpin may qualify for the PIP-binding interface (Fig. 5a, b, Supplementary Fig. 9). To identify the PIP-binding site of the Bcr-Abl PH domain experimentally, we soaked the PH domain crystals with a soluble PI(4,5) $P_2$ derivative and found additional electron density at the expected site (Fig. 5b, Supplementary Fig. 9). This density partially fits the PI(4,5)$P_2$ headgroup, although minor steric clashes remain with the surrounding residues.

We next investigated the PIP-binding selectivity in solution using PIP strips (Fig. 5c). We found that the Bcr-Abl PH domain bound to a wide range of mono, bis- and tris-phosphorylated PIPs, as well as phosphatidylserine, while no lipid binding was observed with the DH domain (Fig. 5c). Mutations in the PIP-binding motif, R723A/K724A or R726A, reduced binding to all interacting PIPs by 60–80% on the PIP strips (Fig. 5a–c). To further quantify these interactions and measure preferences for specific lipids, we employed a more physiological liposome microarray-based assay (LiMA)[40]. In this context, we observed that the Bcr-Abl PH domain bound preferentially to bis- and tris-phosphorylated PIPs over mono-phosphorylated PIPs, as well as preferentially to PS-containing liposomes over other phospho- and sphingo-lipids (Fig. 5d, Supplementary Fig. 9, Supplementary Data 2 and 3). The estimated $K_d$ values for binding to all bis- and tris-phosphorylated PIPs were in the range of 1–5 μM, similar to those reported for other PH domains (Supplementary Fig. 9)[41]. Again, no significant binding was observed for the DH domain alone. Interestingly, the Bcr-Abl DH–PH domain showed the same binding specificity as that of the PH domain alone, but with lower affinity. Finally, the affinity for membranes containing PIPs

was strongly reduced for R726A and almost absent for R723A/K724A (Fig. 5b, d). Overall, these results indicated that the Bcr-Abl PH domain binds to several PIPs with comparable affinity, but broader specificity as compared to other PH domains.

**PH mutations affect Bcr-Abl localization and interactions.** In cells, PH domains are commonly involved in targeting proteins to membranes. Given the high binding affinity of the Bcr-Abl PH domain to PI(4,5)$P_2$, this domain could contribute to Bcr-Abl p210 subcellular localization to the plasma membrane. To test this hypothesis, we investigated Bcr-Abl localization by confocal immunofluorescence microscopy. We analyzed the localization of different Bcr-Abl constructs, including Bcr-Abl p210, PIP-binding-deficient mutants of p210 (R723A/K724A and R726A), as well as p190. Additionally, we combined the PH domain mutations with a mutation disrupting the binding of the F-actin binding domain (FABD; Fig. 1) in the very C-terminus of Bcr-Abl to F-actin (F1100E), which previously was shown to result in strongly reduced co-localization of Bcr-Abl p210 with cortical F-actin[27].

Bcr-Abl p210 showed diffuse, predominantly cytosolic, staining, and quantitative image analysis additionally revealed a strong co-localization with F-actin (Fig. 6a). Both mutations of the PH domain (R723A/K724A and R726A) lead to a small reduction in F-actin co-localization and showed a very similar localization pattern to that of p190 lacking the PH domain. In line with previous results, the FABD mutant F1100E strongly reduced F-actin co-localization. In this context, no additional effect of the PH domain mutations was observed (Fig. 6a, b).

In an independent line of investigation, we assessed the role of the PH domain in p210 protein–protein interaction. We focused on the tyrosine phosphatase Sts1, which we previously found as a prominent p210 interactor that interacted much more weakly with p190[8]. Bcr-Abl immunoprecipitation experiments showed a strongly reduced amount of Sts1 with both PH mutants, as compared to wild-type p210, almost reaching a similar level of Sts1 interacting with p190 that lacks this domain (Fig. 6c, d, Supplementary Fig. 10). Interestingly, the F1100E mutation of the FABD also reduced Sts1 interaction with p210, whereas, in combination with the PH mutation, no additional decrease was observed in line with our immunofluorescence data.

These data show that PH-mediated PIP binding contributes to Bcr-Abl p210 membrane localization, thereby facilitating the incorporation of Sts1 into the Bcr-Abl p210 complex and thus influencing the p210-signaling network. Our findings indicate that PIP binding may contribute to differential signaling networks between the p210 and p190 isoform.

**Discussion**

We have performed an extensive structural and functional characterization of the Bcr-Abl DH–PH domain. With the structures of the DH and the PH domain, the only missing domains of Bcr-Abl were finally structurally characterized and enabled us to study the molecular basis for the signaling contributions of these domains and to construct a preliminary model of the overall Bcr-Abl structure (Fig. 7).

Our structural data show that with its extended α4–α5 loop blocking the DH-GTPase interface and lack of conserved catalytic residues, the Bcr-Abl DH domain may not act as a GEF. In line with these findings, the Bcr-Abl DH domain did not interact with or directly activate the three GTPases RhoA, Rac1, and Cdc42 in our functional experiments. Importantly, in contrast to previous studies[11, 12], our experimental set-up using the isolated and purified DH–PH fragment, together with purified Rho GTPases, allowed us to directly address possible GEF activity of the

DH–PH domains and to exclude possible indirect effects by other GEFs in cells, as well as downstream signaling pathways regulated by any of the other Bcr-Abl domains. We consider GEF activity on other members of the Rho family unlikely, although we cannot fully exclude it at this point. While the Rho GTPase family comprises up to 25 members, RhoA, Rac1, and Cdc42 represent the most studied members of the Rho family and have also been used in previous studies on RhoGTPase activation in Bcr-Abl-expressing cells[10, 42]. In addition recent functional proteomics studies on six Rho GTPases (including RhoA, Rac1, and Cdc42), as well as on the Bcr-Abl interactome from others and us, did not indicate that Rho GTPase directly interact with Bcr-Abl[8, 9, 43]. We also carefully checked the structural location and possible functional role of the S509A mutation that was previously proposed to modulate leukemogenesis in a mouse model[13], but found no supporting evidence for these findings, as the DH domain structure or binding remains unaltered (Supplementary Fig. 2F).

Diverse mechanisms for the regulation of DH-mediated GEF activity have been described, including direct intramolecular interactions with adjacent linkers of the PH domain, allosteric regulation via phospholipid binding to the PH domain, as well as post-translational modifications of the DH domain, such as phosphorylation[10]. Our results do not support any of these regulation mechanisms in the case of the Bcr-Abl DH–PH domain. The high degree of inter-domain flexibility as observed by SAXS speaks against a regulatory impact of the PH domain on the GEF function of the DH domain. Shortening of the α4–α5 loop of the DH domain was not activating in our GEF assays, excluding a possible inhibitory role of this loop that could block the DH-GTPase interaction surface, in addition to the lack of conservation of residues that are important for GTPase interaction in other GEF proteins. Finally, three tyrosine phosphorylation sites in the DH domain, Y554, Y591, and Y644, were repeatedly mapped in the Bcr-Abl DH domain[8, 44, 45]. Whereas Y544 and Y591 are conserved among other DH domains, Y644 is not, but forms part of the conserved region 3 (CR3) that builds part of the GTPase-interaction interface. The biological impact of the phosphorylation of these tyrosines may merit deeper analysis. However, in cases where the regulation upon DH domain phosphorylation has been studied[46], GEF activity was only mildly modulated upon phosphorylation. Collectively, based on our detailed study, there is no evidence that the Bcr-Abl DH domain acts as a RhoGEF. Hence, it may therefore be dubbed 'pseudo-GEF', along with six other Dbl family members that lack key catalytic residues[47]. Still, the DH domain may play a possible important role for the overall structural organization and integrity of Bcr-Abl p210. We postulate that the activation of RhoA that had been observed for Bcr-Abl p210, but not for the Bcr-Abl p190 isoform in two previous studies[11, 12], may be the consequence of the drastically altered signaling networks between the two isoforms rather than the presence of the DH–PH domain in p210.

Our results indicate that the PH domain contributes to the signaling network of Bcr-Abl through PIP-binding. While Bcr-Abl has been studied for decades, and interaction domains with both unmodified (e.g., SH3 and FABD) and post-translationally modified (SH2) protein partners were shown to be critical for localization and signaling, the PIP binding of the PH domain has not been implicated in Bcr-Abl signaling thus far. Here, we show that PIP binding of the Bcr-Abl PH domain has important consequences on the Bcr-Abl signaling network. Given that Bcr-Abl oligomerizes via its N-terminal coiled-coil domain, the lipid-binding affinity of the isolated PH domain determined here likely additionally profits from an avidity contribution of multiple PH domains in the Bcr-Abl tetramer (Fig. 7). Regarding the cellular localization of Bcr-Abl, our microscopy experiments indicated, in

agreement with previous results, that cortical actin binding mediated via the FABD is a dominant factor in Bcr-Abl localization[27]. In line with this observation, the localization of the Bcr-Abl p210 and p190 isoforms does also not differ drastically, as both forms of Bcr-Abl contain the FABD. Consequently, the membrane targeting of the PH domain is not as evident as might be expected and compared to other PH domain examples, such as Btk. Still, given the complex spatial organization and dynamics of signaling complexes in mammalian cells, even globally small differences in protein subcellular localization can have strong consequences in signaling. Moreover, as demonstrated here for the Bcr-Abl/Sts1 interaction, even small differences in the 'microenvironment' of the two Bcr-Abl isoforms lead to a different subset of interacting or substrate proteins modifying the entire signaling network.

Finally, the overall architecture of Bcr-Abl is another important aspect to consider with regard to cellular localization and recruitment of interaction partners. The presence of the DH–PH tandem domain in the p210 isoform contributes a quarter of the size of the whole protein and it is not unlikely that these additional domains strongly affect the overall structural organization of this multi-domain oncoprotein. Currently it is not well understood whether and how the DH–PH module interacts with its neighboring domains or if, in the context of higher oligomers, the DH domain could dimerize or simply act as a long spacer separating the coiled-coil domain from the kinase domain. Furthermore, as monobodies tend to target functional surfaces, it is not unconceivable that the epitopes for the DH and PH monobodies might represent sites of yet undiscovered functional importance[30].

Our study closes the last gap of high-resolution structural data on individual Bcr-Abl domains that have been studied over the past 20 years. The structures of the DH and PH domains now enable future structural on higher order structural organization of multi-domain constructs and full-length Bcr-Abl p210 by SAXS, cryo-EM and crystallography. The results of these studies will be essential to understand the structural differences between Bcr-Abl p210 and p190 and relate them to the Bcr-Abl signaling networks and its interactors.

## Methods

**Protein expression and purification.** The DH–PH (residues 487–893) and DH (residues 487–702) constructs were cloned with an N-terminal His-tag and TEV cleavage site into pETM-11, the PH domain (residues 704–893) and all GTPases were cloned with an N-terminal GST-tag and TEV-cleavage site into the pETM-30 vector. Primer sequences are shown in Supplementary Data 4. Monobodies were cloned into the pHBT1 vector with an N-terminal His- and Avi-tag, followed by a TEV cleavage site[48]. Similarly, the DH–PH and PH constructs for monobody generation were cloned in the pHBT1 vector with an N-terminal Avi-tag for biotinylation. Expression of the DH–PH, DH and PH constructs was conducted overnight in Rosetta(DE3) (Novagen/Merck Millipore catalog no. 70954) in LB medium after induction with 0.5 mM IPTG at an optical density of ~1.0. The temperature during the expression was reduced to 18 °C. Biotinylation of the DH–PH and PH constructs with Avi-tag was achieved upon co-expression with the biotinylase BirA in BL21(DE3) in the presence of 50 μM biotin[49]. The monobodies were expressed in BL21(DE3) using auto-induction medium (Formedium catalog no. AIMLB0210). During overnight expression the temperature was lowered to 20 °C. Finally, the GTPases were expressed overnight at 18 °C in BL21(DE3) using Terrific Broth (TB) medium and 0.5 mM IPTG for induction. The purification buffer consisted of 50 mM Tris-HCl pH 7.5, 500 mM NaCl, 5% glycerol, and 1 mM DTT. For purification of the GTPases the buffers additionally included 10 mM MgCl₂. Lysis of all samples was conducted on an EmulsiFlex-C5 homogenizer (Avestin) using 18,000 PSI output pressure and three lysis cycles. The cleared supernatants were in a first step purified by gravity flow Ni-NTA Agarose (Qiagen) prior TEV cleavage during overnight dialysis (25 mM Tris-HCl, pH 7.5, 150 mM NaCl, 5% glycerol, and 1 mM DTT). For the formation of the monobody complexes (DH/Mb(Bcr-DH_4) and PH/Mb(Bcr-PH_4)), the proteins were mixed after affinity purification with a twofold excess of monobody prior the dialysis and TEV cleavage. TEV protease, the His-tag and un-cleaved protein were removed by a reverse Ni-affinity step. The final purification step for all constructs was a size-exclusion-chromatography step depending on the size of the constructs either on a

Superdex75 or Superdex200 16/600 column (25 mM Tris-HCl, pH 7.5, 150 mM NaCl, 5% glycerol, and 1 mM DTT). The samples were concentrated for the subsequent downstream application by Amicon-Ultra centrifugal filters (Millipore) and all proteins could be stored at −80 °C.

**Monobody generation and initial characterization.** The general approach for monobody generation has been previously reported[29]. Two different monobody phage-display libraries were screened, the loop only library[50] and the loop and side library[29]. A biotinylated DH–PH domain sample in the presence of 25 μg ml−1 poly lysine[51] was used for sorting of phage-display libraries. Four rounds of phage-display library selection were performed at concentrations of 100, 100, 50, and 20 nM of biotinylated DH–PH domain for the first, second, third, and fourth rounds, respectively. After amplification of the cDNA encoding the resulting pool of enriched clones, the sub-library was transferred to the yeast display format, as described previously[29] and selection performed at 20 nM DH–PH or 100 nM of PH domain. Clones were analyzed for target protein binding in the yeast-display format and sequenced. Binding assays in the yeast display format were performed as previously described[29] with a concentrations range of the biotinylated targets from 10 nM to 1 μM in Tris-buffered saline (TBS) with 0.1% BSA. Incubation of target protein, yeast cells, and a mouse anti-V5 antibody were for 30 min at RT. Streptavidin-DyLight650 and a FITC-coupled antimouse IgG was added after two washing steps and incubated for 30 min at RT. Samples were analyzed on a Gallios (Beckman Coulter) or BD Accuri C6 (BD Bioscience) flow cytometer. Plots of the mean fluorescence intensity vs. target concentration and fitting a 1:1 binding model to the data using Prism (GraphPad) was used to determine the apparent $K_d$ values. We have established that the apparent $K_d$ values determined using this method generally agrees with those determined from biophysical measurements of purified proteins[29].

**Isothermal titration calorimetry.** Isothermal titration calorimetry (ITC) measurements were performed on a MicroCal iTC200 (GE) instrument. All proteins were degassed after extensive dialysis against 25 mM HEPES (pH 7.5), 150 mM NaCl, 0.5 mM TCEP. Protein concentrations were determined by measuring UV absorbance at 280 nm. The titrations were performed in 16 steps with 0.49 μL injections for the first and 2.49 μL for the other injections. Concentrations were selected based on assumed affinities and the signal strength of the interaction. The MicroCal software was used for data analysis.

**NMR spectroscopy.** For NMR structural studies, the DH domain was expressed in minimal M9 medium supplemented with 1 g L−1 15NH4Cl and either 2 g L−1 13C-glucose (Cambridge Isotope Laboratories) or 4 g L−1 unlabeled glucose. Finally, the samples were dialyzed into an NMR compatible buffer containing 50 mM HEPES, pH 7, 50 mM NaCl, 1 mM DTT. For GTPase titration experiments, the buffer additionally contained 1 mM imidazole and 2 mM EDTA.

For the NMR structure determination, uniformly 15N, 13C-labelled DH domain (487–702) was prepared at a concentration of 850 μM. Measurements for the assignment were carried out at 303 K on Bruker Avance spectrometers equipped with cryogenic 1H{13C,15N} probes operating at proton resonance frequencies of 600–950 MHz. Backbone assignment was achieved using TROSY-type[52, 53] BEST[54, 55] versions of HNCO, HN(CA)CO, HNCACB, HN(CO)CACB, and HN(CA)HA triple resonance experiments. Side chain resonances were assigned based on three-dimensional, TROSY-type versions of the H(CCCO)NH-TOCSY and (H)C(CCO) NH-TOCSY, a 3D 13C-start CCH-TOCSY, as well as 3D 13C-separated NOESY experiments. Aromatic side chains were assigned by the following 2D experiments: H(N)CD (Trp), H(NCDCG)CB (Trp), (HB)CB(CGCD)HD, (HB)CB(CGCD-TOCSY)Har. NOE distance restraints were derived from 15N and 13C-separated NOESY spectra. In addition to the manual assignment strategy, we employed the FLYA automated resonance assignment algorithm for backbone and side chain assignment[56]. The 1H/13C/15N resonance assignments were complete for all CH_n groups and the backbone NH groups. About 90% of these assignments (from FLYA) were verified manually. Automated NOE assignment and structure calculation was conducted in CYANA[57]. The structure calculation was based on 4298 NOE upper distance limits, of which 845 were long-range. Structure calculations with torsion angle dynamics were started from 200 random conformers[58]. A consensus structure bundle of 20 conformers was subjected to restrained energy minimization in explicit solvent against the AMBER force field with OPALp[59–61]. RMSD values were calculated for the well-defined regions of the protein, residues 496–619, 638–693, as determined with CYRANGE[62].

The interaction studies with the GTPases were done using a uniformly 15N-labeled DH domain at a concentration of 50 μM. GTPase stock solutions ranged between 600 and 900 μM. All proteins were dialyzed into the same buffer prior to the experiment. Chemical shift changes were monitored using [15N,1H]-TROSY experiments recorded on a Bruker Avance spectrometer with cryogenic 1H {13C,15N} probe operating at a proton resonance frequencies of 800 MHz. The composite 15N and 1H chemical shift perturbation was calculated with a scaling factor for the 15N shift changes of 0.14[63]. Binding constants were fitted in Prism (GraphPad) according to a 1:1 binding model.

**X-ray crystallography.** Samples for crystallization trials were concentrated to ~10 mg mL−1. Initial screenings were set-up in a 96-well plate with sitting drop crystallization using the Mosquito robot (TTP Labtech). Drops were composed of a mixture of protein and buffer in a 1:1 ratio (v/v) and crystals grown at 291 K. The DH/Mb(Bcr-DH_4) complex crystallized in condition F5 of the Morpheus HT screen (MD1-47, Molecular Dimensions) corresponding to 0.12 M monosaccharides (0.2 M D-glucose; 0.2 M D-mannose; 0.2 M D-galactose; 0.2 M L-fucose; 0.2 M D-xylose; 0.2 M N-acetyl-D-glucosamine), 0.1 M buffer system 2 pH 7.5 (sodium HEPES; MOPS (acid)) and 50% precipitant mix 1 (40% v/v PEG 500 MME; 20% w/v PEG 20000). The PH/Mb(Bcr-PH_4) complex crystallized in condition 81 of the JCSG+ suite (Qiagen) that contained 0.1 M potassium thiocyanate and 30% (w/v) PEG MME 2000. PH/Mb(Bcr-PH_4) crystals could be improved in the presence of 20% glycerol.

X-ray diffraction data were collected at the synchrotron SLS (Swiss Lightsource, Villingen, Switzerland), beamline ×06DA using a single wavelength at 100 K. The diffraction data were processed with the X-ray Detector Software (XDS, http://xds.mpimf-heidelberg.mpg.de/). Phasing was achieved by molecular replacement in Phaser using homology models created in SwissModel[64, 65]. The homology model for the DH and the PH domain was based on the template with PDB ID 2Z0Q, the monobody was modeled based on template PDB ID 3TEU. Manual model building, solvent addition and refinement employed phenix.refine in the Phenix software suite[66].

**Small-angle X-ray scattering.** Proteins for SAXS analysis were concentrated to 10–15 mg mL−1 and dialyzed into a SAXS compatible buffer (25 mM Tris-HCl pH 7.5, 150 mM NaCl, 5% glycerol, 1 mM DTT). SAXS data were collected at EMBL P12 beamline, DESY, Hamburg, Germany with protein concentration ranges of 1.6–14.4 mg mL−1. The data were cropped from the first point of the Guinier region until the end of the useful range defined by SHANUM[67]. High- and low- concentration curves were merged to counter concentration effects, such as interparticle interference using the program PRIMUS from the ATSAS package[68]. GNOM was used to obtain the $p(r)$ and determine the corresponding $D_{max}$ and $R_g$ values[69]. SREFLEX was used to generate and refine high-resolution hybrid models using high-resolution structures as the starting point[34]. EOM was used to generate an ensemble of conformations and to assess the inter-domain flexibility of the DH–PH domain[35]. The scattering curves from the high-resolution models were calculated using CRYSOL[70]. SAXS figures were prepared using PyMOL (www.pymol.org) and SASpy[71].

**Bioinformatics tools.** Sequence and secondary structure analysis relied on webservers for BLAST (blast.ncbi.nlm.nih.gov), PSIPRED (bioinf.cs.ucl.ac.uk/psipred) and HHpred (toolkit.tuebingen.mpg.de/#/tools/hhpred)[72–74]. Homology models were created using SwissModel (swissmodel.expasy.org)[64]. Structural analysis and figure preparation was conducted in PyMOL (www.pymol.org).

**GTPase interaction studies.** DH–PH and DH constructs were mixed with GTPases in a 1:2 molar ratio and dialyzed into either EDTA- or MgCl2-containing buffer (25 mM Tris-HCl, pH 7.5, 150 mM NaCl, 2 mM DTT, and 1 mM EDTA or 10 mM MgCl2) overnight at 4 °C. Half of the sample was then directly loaded onto an analytical Superdex75 10/300 size-exclusion column. For pull-down analysis, the samples were twofold diluted with binding buffer (25 mM Tris-HCl, pH 7.5, and 150 mM NaCl), mixed with Ni-NTA Sepharose (GE Healthcare) and incubated for 2 h at 4 °C. The slurry was transferred into a micro spin column and the unbound fraction was collected by centrifugation at 500×g for 1 min. The resin was washed with a total of 10 column-volumes buffer (25 mM Tris-HCl, pH 7.5, 150 mM NaCl, 1 mM DTT, 10 mM imidazole) prior to elution of the bound proteins in 3 column-volumes buffer (25 mM Tris-HCL, pH 7.5, 150 mM NaCl, 1 mM DTT, and 500 mM imidazole). The samples were analyzed on Coomassie-stained SDS-PAGE. Equal amounts of input, flow-through, wash and elution fractions were analyzed on SDS-PAGE, scanned with the Odyssey imager (Li-Cor) and quantified using the Image Studio Software (Li-Cor). The percentage of untagged protein in the respective gel lane normalized to the input fraction was calculated.

**Nucleotide exchange assay.** GTPases were dialyzed in loading buffer (20 mM Tris-HCl, pH 7.5, 100 mM NaCl, 5 mM EDTA, 1 mM DTT, 5% glycerol) overnight at 4 °C to remove bound nucleotide. The nucleotide-free GTPases were then mixed with a fivefold excess of mant-GDP (Thermo Fisher Scientific) and incubated for 90 min on ice. The mant-GDP-loaded GTPases were stabilized upon addition of 10 mM MgCl2 with an additional incubation time of 30 min on ice. Excess nucleotide was then removed with a PD-10 Desalting column (GE Healthcare) and the protein concentration determined by absorbance measurement.

Nucleotide exchange reactions were set-up in a black 96-well plate with 1 μM GTPase/mant-GDP and 10 μM GTP in reaction buffer (20 mM Tris-HCl, pH 7.5, 100 mM NaCl, 1 mM DTT, 5% glycerol). The exchange reaction was started upon addition of 20 mM EDTA or 0.5 μM Dbs DH–PH as positive controls, and 5 μM or 50 μM Bcr-Abl DH–PH and DH domains and immediately measured at the SpectraMax M5 plate reader (Molecular Devices). The single-wavelength measurement was done at 25 °C with an excitation at 360 nm and emission at 445 nm. The decrease of the fluorescence upon GDP-to-GTP exchange was monitored

up to 20 min, when the intrinsic exchange reached the baseline. The exchange rate constants were fitted in Prism (GraphPad) and the enhancement factor was calculated in comparison to the intrinsic exchange rate without addition of EDTA or GEF protein.

**PIP strips lipid-binding assay**. PIP strips (Echelon Biosciences) were blocked in PBS-T + 3% BSA for 1 h at room temperature before adding 20 nM of the target proteins for another 1 h at room temperature. Membranes were then washed three times in PBS-T and incubated with an anti-His antibody (Qiagen) in PBS-T + 3% BSA overnight at 4 °C. After washing again three times in PBS-T, detection was achieved with a secondary antimouse antibody coupled to an IRDye800 (1 h, room temperature) on the Li-Cor Odyssey imaging system (Li-Cor Biosciences) and quantification was done in the Image Studio software (Li-Cor Biosciences).

**LiMA experimental procedure**. The fabrication of liposome microarrays, the experimental procedure of protein-liposome interaction assay, the image analysis, and the calculation of normalized binding intensities (NBIs) values were described previously[40]. Briefly, liposomes were formed in the assay buffer (10 mM HEPES, pH 7.4, 150 mM NaCl) from lipid mixtures containing various combinations of 17 signaling lipids (Supplementary Data 2). The liposomes were incubated 20 min with different concentrations of purified sfGFP-tagged proteins or cell extracts containing sfGFP-tagged proteins with known lipid-binding preferences (control proteins), prepared as described previously (Supplementary Fig. 8)[40]. Subsequently, the unbound material was washed and the interactions were monitored by automated microscopy. Fluorescence intensities from pixels matching liposomal membranes were extracted and used for calculation of normalized binding intensities (NBI).

**Immunofluorescence microscopy**. HeLa cells were transfected with vectors encoding p210, mutants thereof or p190 under the control of an SV40 promoter by using PEI transfection reagent (polyethylenimine, Polysciences). After 48 h of incubation, the medium of transfected cells was removed and the cells were quickly washed with phosphate buffer saline (PBS) equilibrated at 37 °C. Coverslips were then incubated in 3% paraformaldehyde (PFA). After 15 min incubation on ice, the cells were washed three times 5 min with PBS. The cells were permeabilized for 10 min at room temperature with PBS + 0.5% Triton X-100 and blocked with a commercial 10% normal goat serum blocking solution (Invitrogen 50-197Z) for 10 min. Incubation with the primary antibody solution (pAb mouse anti-HA.11, Covance, dilution 1:1000 in PBS or pAb mouse anti-myc, Cell Signaling, dilution 1:8000 in PBS) was performed for 1 h and followed by three washes with PBS + 0.1% Triton X-100. Then, the secondary antibody solution (goat antimouse Alexa-488 from Thermo Scientific, R37120, dilution 1:1000 in PBS) was applied for 30 min. After application of the secondary antibody, all incubation steps were performed protected from light. After washing, the actin cytoskeleton was stained by incubation of the cells with phalloidin-rhodamin for 20 min according to supplier instructions (Thermo scientific, R415, phalloidin-rhodamin resuspended in methanol and dilution: 5 µL methanolic stock solution into 200 µL PBS + 1% BSA). After three times 5 min washes in PBS + 0.1% Triton X-100, DNA was stained with Hoechst for 10 min (dilution 1 µg mL$^{-1}$ in PBS). Finally, three times 5 min washes in PBS + 0.1% Triton X-100, coverslips were mounted on a microscopic slide using Mowiol. Images were obtained on a Zeiss LSM780 confocal microscope using a Plan-Apochromat 63x/1.40 Oil DIC M27.

**Actin co-localization**. All images were acquired with the same confocal microscope settings (74 nm pixel size, 355 nm z-plane spacing, 489 nm laser excitation and 533 nm detection band for Alexa-488 immunostained constructs, 566 nm laser excitation and 634 nm detection band for phalloidin-rhodamin stained actin, with same laser powers and detector gains for all samples). Cells with similar shape and actin morphology were selected for all conditions. For co-localization of Bcr-Abl with actin, we computed the Manders coefficient of co-localization[75] using the ImageJ software, as follows: we background-corrected the Bcr-Abl channel (Bcr-Abl total) and applied a threshold on the actin channel; next we used the thresholded actin image to mask the background-corrected Bcr-Abl channel (Bcr-Abl actin); image stacks from 'Bcr-Abl total' and 'Bcr-Abl actin' were sum-projected along the z-axis. The fraction of Bcr-Abl co-localizing with actin was calculated by dividing the sum intensity of 'Bcr-Abl actin' by the sum intensity of 'Bcr-Abl total' in a manually drawn ROI around each cell in the projected images.

**Bcr-Abl immunoprecipitation**. A detailed description of the Bcr-Abl immunoprecipitation was described previously[8]. Bcr-Abl isoforms and mutants were expressed in murine BaF3 cells (DSMZ no. ACC-300) and cell lysis and purification was performed in TAP buffer (50 mM Tris-HCl, pH 7.5, 100 mM NaCl, 5% glycerol, 0.2% (w/v) NP-40 Alternative) containing protease and phosphatase inhibitors (25 mM sodium fluoride, 1 mM vanadate, 1 mM PMSF, 10 µg mL$^{-1}$ TPCK, 1X Complete protease inhibitor (Roche). Normal mouse IgG antibody (Thermo Scientific, catalog no. 10400C) and the monoclonal Abl antibody (clone 24–21)[76] were covalently coupled to NHS-activated sepharose (GE Healthcare). The lysates were pre-cleared for 1 h at 4 °C with the IgG-NHS resin and the

supernatant then transferred to the Abl-NHS resin for another 3 h at 4 °C. Elution fractions containing Bcr-Abl were pooled.

**Data availability**. Coordinates have been deposited in the Protein Data Bank with the accession codes: 5N6R (DH NMR structure), 5N7E (DH/Mb(Bcr-DH_4) and 5OC7 (PH/Mb(Bcr-PH_4)). NMR chemical shift assignments are deposited in BMRB entry 34101. The collected SAXS data and the generated high-resolution hybrid models have been deposited at SASBDB (entry SASDC26, SASDC36, SASDC46)[77]. Other data are available from the corresponding author upon reasonable request.

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

## Acknowledgements

This work was supported by the ISREC Foundation (S.R., T.K., S.G., S.P., and O.H.), Swiss National Science Foundation (grant 31003A_140913; S.R. and O.H.), National Center of Competence in Research (NCCR) in Chemical Biology (O.H. and T.K.), European Research Council (Grant ERC-2016-CoG 682311-ONCOINTRABODY to O.H.), the US National Institutes of Health (R01-GM090324 to S.K.). C.G. and A.P. were supported by the EMBL Interdisciplinary Postdoc Programme under Marie Curie COFUND Actions. The NMR projects were supported by BioNMR (contract no. 261863) and iNEXT (contract no. 653706). We thank the PSI, Villigen, Switzerland for synchrotron radiation beamtime at beamline ×06DA of the Swiss Light Source, as well as the EMBL BioSAXS P12 beamline and acknowledge their support with data collection. We thank A. Lamontanara and D. Duarte for their support with SAXS measurements, C. Tischer from the Advanced Light Microscopy Facility (ALMF) at the European Molecular Biology Laboratory (EMBL) in Heidelberg for support in microscopy and image analysis. We also thank G. Mann and T. Reichart for their critical input on the manuscript, and all members of the Hantschel lab for continuous support and discussions.

## Author contributions

S.R., F.L., A.R., V.D., and F.P. conducted and analyzed experiments in Fig. 2. A.P. and D.S. conducted and analyzed experiments in Fig. 3. S.R., C.G., and A.-C.G. conducted and analyzed the experiments in Figs. 5, 6. S.R. and D.H. conducted and analyzed experiments in Fig. 4. S.G. and S.P. contributed experiments to Fig. 6. B.G. contributed experiments in Fig. 5b. S.G. provided technical assistance and vital tools for all experiments. L.B. and P.G. provided important bioinformatics support for NMR structural data analysis. T.K., A.K., and S.K. designed and conducted monobody generation and analysis. S.R. and O.H. coordinated this study, designed the experiments, interpreted the data and wrote the manuscript.

## Additional information

**Competing interests:** The authors declare no competing financial interests.

