## [Peer Review File · Nature Communications]

Reviewers' comments:

Reviewer #1 (Remarks to the Author):

The study by Reckel and colleagues is aimed at structural characterization of two domains of the tyrosine kinase Bcr-Abl. The authors determined the solution structure of the DH domain and separately the crystal structure of the PH domain and tested binding of these domains to potential binding partners. Overall, the manuscript reports on the new finding: surprisingly the atomic resolution structures of the DH and PH domains of Bcr-Abl have not been determined in spite the well-documented role of Bcr-Abl in oncogenesis and in spite the fact that numerous structures of both PH and DH domains from other proteins are available. However, the structural part needs to be accompanied by a large part describing novel functional or biological aspects that is absent in the current manuscript.

Furthermore, the manuscript requires substantial rewriting and restructuring in order to highlight the most interesting results upfront rather than showing data in a more chronological order. The first three pages of the Results section should be condensed into a couple of sentences as they essentially describe preparation of samples. It is unclear as to why the authors thoroughly investigate binding of monoclonal antibody since it was used as a reagent for crystallization. The conclusion that the DH and PH domains do not influence each other needs to be confirmed by a reverse titration of unlabeled DH to the 15N-labeled PH domain (page 12). There is no evidence that the PH domain was properly folded when it was added to the 15N-labeled DH domain sample. Equally important is to confirm the integrity and structural stability of the DH-PH domain construct before drawing the strong conclusion that the DH-PH region does not exhibit the RhoGEF activity. Binding to PIs by DH-PH should be included and compared to binding by the isolated PH domain. The structure of the putative complex with PIP2 cannot be reported because of the poor electron density. The binding of the PH domain to PIs is not something surprising or new and does not help much - a more sophisticated (than PIP strip) assay has to be run here. The sentence on page 19 "in contrast to previous studies..." needs citation.

Reviewer #2 (Remarks to the Author):

In this manuscript the structure of the DH and PH domain of the tyrosine kinase Bcr-Abl and their function within the kinase are investigated using a variety of structural and functional techniques. The authors combine different structural techniques (NMR, Xtal, SAXS) to determine the structural arrangement of the DH-PH domain as well as an array of techniques for functional characterization. Overall the manuscript is clearly and concisely written and with a few minor revisions and comments (see below) I support the publication of this manuscript).

In the structural characterization of the DH domain the authors show nicely that the use of monoclonal antibodies to aid crystallization did not significantly alter the structure of the domain. A similar treatment for the PH domain however is missing. The authors should at least comment on the relevance of the presented crystal structure of the PH domain. In particular also in view of fitting the SAXS data of the isolated PH domain, where a homology model was actually used as a starting point for the SREFLEX program.

Generally the SAXS analysis is done using state of the art techniques and programs from the newest ATSAS suite. The programs are employed expertly to the analysis of the data for the isolated domains and the combined DH-PH construct. A few technical points however should be addressed and commented on before publication.

While the author do supply a table with the most relevant SAXS analysis parameters in the

supplementary information, they do not present the Guinier plots for measured data. In order to allow the reader better judge the data quality these should be provided for all the SAXS data collected. Also the different $P(r)$ functions determined from the SAXS data should be depicted. This could also be done in the supplemental information.

The display of the four different conformations of the DH-PH domains determined by EOM should be improved. It is hard to discern the different shades of green especially for the unstructured linker and loop regions. A more schematic representation might be better suited for that.

The errors on the Table 3 in the SI should be double checked for typos (e.g. the error given for the Porod volume of the Dh-PH domain and the error on the molecular mass determined from it are not consistent)

Reviewer #3 (Remarks to the Author):

BCR and its paralog ABR are quite large proteins with diverse domains and motifs, among them a controversial RHOGEF module and the more RAC-specific RHOGEF domain. The functions of these two RHO regulatory proteins are widely unclear. In contrast, the BCR-ABL fusion protein, which generated by a reciprocal translocation between chromosomes 22 and 9 often found in CML patients, has been extensively studied. Among different BCR-ABL proteins, p190 and p210 are the most common gene products, both of which have lost the C-terminal RHOGEF domain of BCR. Unlike p190, p210 has got residues 427-927 of BCR, encompassing the canonical structural motifs of the Dbl family RHOGEF proteins, the so-called DH-PH tandem domains, usually responsible for acceleration of the nucleotide exchange reaction of RHO proteins.

The submitted manuscript describes structural and biochemical characterization of BCR DH and PH domains, respectively. The authors' success in exploiting different biophysical methods and tools for the determination of the DH and PH structures is rather encouraging. Functional analysis have shown that DH does not exhibit any GEF activity neither for non of the analyzed RHO proteins, such as RAC1, CDC42, or RHOA. An assumed sterical clash with binding of RHO proteins by an extended $\alpha 4$ - $\alpha 5$ loop at the edge of the DH domain could unfortunately not be proved neither for the DH nor for the DH-PH proteins. Further analysis have shown that there is no physical interaction between both DH and PH domains. Functional studies on PH domain-membrane interaction have indicated that this domain shows a broad selectivity for phosphoinositides, and is responsible for p210 membrane association, most probably facilitated by F-actin and tyrosine phosphatase STS1.

I believe that the time and care that the authors have invested in this study are remarkable and exemplary. This manuscript is clear and well described. It adds, however, little new insights on understanding the DH domain function for p210. There is a large list of reports with structural and functional investigations on the DH-PH domains of the DBL family. There are meanwhile more than 60 and 190 structures of various DH and PH domains, respectively, available in the database. A lack of GEF activity of the DH domain of BCR and six other DBL family proteins has previously been claimed by Jaiswal et al., JBC (2013). Notably, obtained data are also valid for the BCR proteins itself. Exploring alternative functional role(s) for the DH domain will be very interesting in providing novel mechanistic clues about BCR-ABL function and eventually disease pathogenesis.

Reza Ahmadian

Reviewer #4 (Remarks to the Author):

Review Sept 15-2017

Structural and functional dissection of the DH and PH domains of oncogenic Bcr-Abl tyrosine kinase.

The authors structurally and functionally characterize the DH and PH domains of Bcr-Abl p210. Differences in phosphorylation and protein interactions between p210 and p190 (a shorter isoform lacking the DH-PH region) have been previously noted, thus the interest in investigating these tandem domains.

Overall the presented work is of high quality and the conclusions drawn are very reasonable. I recommend publication with only minor modifications.

The DH domain structure was determined by NMR and by crystallography in complex with a monobody. The PH domain structure was also solved by crystallography in complex with a monobody. SAXS analysis indicated the DH and PH domains in p210 exist in an extended conformation, consistent with their NMR results where chemical shifts of the DH domain were not observed upon addition of the PH domain.

Interaction of RhoA, Rac1 and Cdc42 were not detected with DH and DH-PH protein fragments by pull-down or co-migration through sizing column while Dbs DH-PH served as a positive control. NMR analysis of the Bcr-Abl DH domain revealed a low affinity interface for all three GTPases that mapped to the canonical GTPase binding site. Nucleotide exchange activity of the p210 DH or DH-PH regions was not detected for the three GTPases tested however. The authors declare that Bcr-Abl DH-PH is a pseudo-GEF consistent with a lack of functionally conserved residues in the DH domain sequence compared to other RhoGEFs.

The Bcr-Abl PH domain was shown to bind several phosphoinositide that were reduced by point mutations of basic residues in the expected PIP-binding pocket. Localization of p210 in cells was shown to revert to that of p190 (lacking the DH-PH domain) upon introduction of the PH domain mutations. Co-IP of Bcr-Abl with Sts1 was reduced to the level seen with p190 by these same PH domain mutations in p210.

The authors conclude that the PH domain is primarily responsible for the differential functional properties of p190 compared to p210 based on the PH domain localization role.

Overall, the authors also provide compelling evidence that the purified DH-PH region of Bcr-Abl p210 is a non-functional GEF.

One minor point the authors might comment on in their discussion:

The authors elude to work published by others that RhoGEF activity of Bcr-Abl p210 has been detected and is impaired by a point mutation (S509A). In the absence of detectable GEF activity with purified protein, how do the authors explain the discrepancy with the prior results? With a structure in hand, can an alternate effect of the S509A mutation be predicted (other than loss of GEF activity)?

Other items to note:

1) Sequence alignment of DH domains would be helpful to demonstrate that residues important for GTPase interaction, catalysis and selectivity are not conserved in Bcr-Abl p210 compared to other functional Rho GEFs. A sequence alignment of the DH and PH domains would also serve to highlight

the loop insertions and various mutations mentioned within the text.

2) Are there any other DH domain proteins known without GEF activity? Is this the first pseudo-GEF identified?

Point-by-point response to Referees comments:

Reviewer #1 (Remarks to the Author):

The study by Reckel and colleagues is aimed at structural characterization of two domains of the tyrosine kinase Bcr-Abl. The authors determined the solution structure of the DH domain and separately the crystal structure of the PH domain and tested binding of these domains to potential binding partners. Overall, the manuscript reports on the new finding: surprisingly the atomic resolution structures of the DH and PH domains of Bcr-Abl have not been determined in spite the well-documented role of Bcr-Abl in oncogenesis and in spite the fact that numerous structures of both PH and DH domains from other proteins are available. However, the structural part needs to be accompanied by a large part describing novel functional or biological aspects that is absent in the current manuscript.

We welcome the positive evaluation and surprise of the reviewer that despite the immense efforts of the Bcr-Abl community and the numerous DH and PH domain structures that were solved, we were the first to solve the DH-PH domain structures of Bcr-Abl. In our opinion, this underlines the difficulty to produce both domains recombinantly and the surprising structural features of both domains, as well as emphasizes the power of using monobodies as crystallization chaperones. We also agree with the reviewer that nowadays structural studies need to be accompanied by data to study biological function. In fact, such data is abundantly present in our manuscript. Figures 4, 5 and 6 contain extensive and in-depth biological data including various biochemical binding studies, enzymatic assays, large-scale protein-lipid arrays, quantitative immunofluorescence microscopy and protein interaction studies in cells of both domains in isolation, as tandem domains and in the context of the full-length Bcr-Abl protein. Therefore, we believe that these results provide coherent and comprehensive structural and functional evidence supporting the conclusions of our manuscript.

Furthermore, the manuscript requires substantial rewriting and restructuring in order to highlight the most interesting results upfront rather than showing data in a more chronological order. The first three pages of the Results section should be condensed into a couple of sentences as they essentially describe preparation of samples.

We carefully checked the composition and content of the Results section and respectfully disagree with the reviewer. The first 3.5 pages of the Results section contain the entire description and structural analysis of the NMR structure of the DH domain, the crystal structure of the DH domain, as well as the crystal structure of the PH domain. These are the major results of our paper on which all further biological experiments are based on. Furthermore, we strongly feel that we should present the readers of our paper a logical (in this case also chronological) order of our experimental strategies, approaches and results, as long as this is in line with the length restrictions of the journal.

It is unclear as to why the authors thoroughly investigate binding of monobody since it was used as a reagent for crystallization.

We measured the binding affinities of the selected DH- and PH-domain targeting monobodies using ITC (SI Fig. 3). Our scientific rigor and aim to exclusively use well-

characterized tools for all our experiments mandate to always characterize binding affinities of any new monobody that we obtain after selection. Even for the action as a crystallization chaperone, it is important to know the precise binding affinity of a monobody, because high-affinity monobodies are more likely to bind to a highly populated conformation (see also our response to a comment by Reviewer #2). Also, there was the chance that the monobodies also might be acting as competitive inhibitors of either the DH- or PH- domain. As is the case for monobodies that we have reported in the past, monobodies are powerful tools for future experiments. Hence, we strive to provide as comprehensive information as possible. The reviewer also may take into consideration that the data is only shown in the SI and only very briefly mentioned in the main text. Therefore, we think that this data is not overly distracting the flow of the paper.

The conclusion that the DH and PH domains do not influence each other needs to be confirmed by a reverse titration of unlabeled DH to the ¹⁵N-labeled PH domain (page 12).

We thank the reviewer for this suggestion. We attempted to use the PH domain in NMR studies, but the ¹⁵N-HSQC spectrum showed little dispersion, which is likely due to the 59-aa insertion. This would have hampered or prevented the interpretation of the proposed experiment. In addition, the reverse titration experiment would not provide additional insight, as it could only further back up clear-cut (negative) data from the (forward) NMR titration and the SAXS analysis. Overall, the finding that the DH and the PH domain do not interact with each other is also not an important conclusion of the paper, interactions of DH- and PH-domains were only observed in few DH-PH units.

There is no evidence that the PH domain was properly folded when it was added to the ¹⁵N-labeled DH domain sample. Equally important is to confirm the integrity and structural stability of the DH-PH domain construct before drawing the strong conclusion that the DH-PH region does not exhibit the RhoGEF activity.

We did thorough quality checks using Size Exclusion Chromatography coupled to Multiangle Light Scattering (SEC-MALS), Far-UV CD spectroscopy + melting curves, as well as 1D- and 2D-NMR experiments for the recombinant proteins that were used in all crystallography, NMR, SAXS, as well as all binding and enzymatic assays (see SI Fig. 1). The results from these experiments all support that the PH domain samples are properly folded. In particular the SAXS reconstruction of the full-length PH domain would not be possible if the PH domain was not properly folded, as SAXS is exquisitely sensitive to aggregation and even small fractions of aggregates become immediately apparent in SAXS experiments and prevent their interpretation. Likewise, it would be highly unlikely to obtain strongly diffracting crystals or strong PIP binding, if the PH domain protein would not be properly folded.

Binding to PIs by DH-PH should be included and compared to binding by the isolated PH domain.

The requested data is shown in Figure 5D and binding of PIPs to PH versus DH-PH is described and discussed on the results sections (lines 349-350 of original manuscript).

The structure of the putative complex with PIP2 cannot be reported because of the poor electron density.

While we agree with the reviewer that only poor electron density can be identified for the PIP2 derivative that was soaked into PH domain crystals, we respectfully disagree to remove the structure for the following reasons. First, our findings on the PIP binding site are reported very cautiously. Secondly, the proposed structural findings are confirmed by PIP binding assays with mutant PH proteins and coincide with the structurally conserved canonical PIP binding pocket of PH domains. Third, we also looked at the crystal data without PIP and the PH domain structure is identical. Fourth, there are dozens (if not hundreds) of published crystal structures, including structures of PH domains, in which the electron density of a lipid (or other) ligand is not as flawless as one would have hoped.

The binding of the PH domain to PIs is not something surprising or new and does not help much- a more sophisticated (than PIP strip) assay has to be run here.

While we agree that 'binding of the PH domain to PIs is not something surprising', the PH domain of Bcr-Abl has never been characterized before and Bcr-Abl is a central oncoprotein and drug target. Therefore, our findings are 'new', as they implicate the PH domain in Bcr-Abl localization, which is important for its oncogenicity. Furthermore, it is important to thoroughly characterize PIP binding selectivity and affinity, as there are strong differences within the PH domain family. Therefore, we have indeed chosen a 'more sophisticated (than PIP strip) assay' that the reviewer must have overseen: Figure 5D, as well as SI Figure 9 report on a more physiological and quantitative liposome microarray-based assay (LiMA) that was developed by our co-author Anne-Claude Gavin and represents an extensively validated and robust assay.

The sentence on page 19 "in contrast to previous studies..." needs citation.

We thank the reviewer for this remark. We have added the respective references (page 20).

Reviewer #2 (Remarks to the Author):

In this manuscript the structure of the DH and PH domain of the tyrosine kinase Bcr-Abl and their function within the kinase are investigated using a variety of structural and functional techniques. The authors combine different structural techniques (NMR, Xtal, SAXS) to determine the structural arrangement of the DH-PH domain as well as an array of techniques for functional characterization. Overall the manuscript is clearly and concisely written and with a few minor revisions and comments (see below) I support the publication of this manuscript).

We thank the reviewer for her/his enthusiasm for our manuscript.

In the structural characterization of the DH domain the authors show nicely that the use of monobodies to aid crystallization did not significantly alter the structure of the domain. A similar treatment for the PH domain however is missing. The authors should at least comment on the relevance of the presented crystal structure of the PH domain. In particular also in view of fitting the SAXS data of the isolated PH

domain, where a homology model was actually used as a starting point for the SREFLEX program.

We thank the reviewer for this suggestion. While we cannot formally exclude that the monobody and deletion of the unique beta5-beta6 insertion may change the PH conformation, the Bcr-Abl PH domain possesses a canonical PH-domain fold and hence has high structural similarity to the Sos1 and other PH domains. This is shown in SI Figure 2e and discussed in the text. Furthermore, from our previous work on monobodies targeting various SH2 domains (see e.g. PDB entries 5MTM, 5MTJ, 5MTN, 3K2M, 3T04, 4JE4, 4JEG, 5DC4) we did not find that monobody binding would significantly alter the structure of its target domain.

Monobodies and antibodies have extensively used as crystallization chaperones. There have been few, if any, cases among these studies that crystallization chaperones "distort" the structure of their targets. The reason is quite simple. The processes of generating crystallization chaperones selectively enrich reagents that bind to one (or more) of low-energy conformations within the native ensemble of their targets. This important but often overlooked characteristic of crystallization chaperones has been elaborated in ref #31, which we now emphasize in the manuscript (p 8).

Generally the SAXS analysis is done using state of the art techniques and programs form the newest ATSAS suite. The programs are employed expertly to the analysis of the data for the isolated domains and the combined DH-PH construct. A few technical points however should be addressed and commented on before publication.

We appreciate the positive evaluation of the reviewer.

While the author do supply a table with the most relevant SAXS analysis parameters in the supplementary information, they do not present the Guinier plots for measured data. In order to allow the reader better judge the data quality these should be provided for all the SAXS data collected. Also the different P(r) functions determined for from the SAXS data should be depicted. This could also be done in the supplemental information.

We have added both the Guinier plots and P(r) functions to SI (new SI Figure 5)

The display of the fours different conformations of the DH-PH domains determined by EOM should be improved. It is hard to discern the different shades of green especially for the unstructured linker and loop regions. A more schamatic representation might be better suited for that.

We apologize for the suboptimal graphical quality of this figure. We have changed the coloring scheme in Fig. 3C to easier discriminate the four depicted conformations.

The errors on the Table 3 in the SI should be double checked for typos (e.g. the error given for the Porod volume of the Dh-PH domain and the error on the molecular mass determined from it are not consistent)

We have checked this and revised SI Table 3. We apologize for the oversight.

Reviewer #3 (Remarks to the Author):

BCR and its paralog ABL are quite large proteins with diverse domains and motifs, among them a controversial RHOGEF module and the more RAC-specific RHOGAP domain. The functions of these two RHO regulatory proteins are widely unclear. In contrast, the BCR-ABL fusion protein, which generated by a reciprocal translocation between chromosomes 22 and 9 often found in CML patients, has been extensively studied. Among different BCR-ABL proteins, p190 and p210 are the most common gene products, both of which have lost the C-terminal RHOGAP domain of BCR. Unlike p190, p210 has got residues 427-927 of BCR, encompassing the canonical structural motifs of the Dbl family RHOGEF proteins, the so-called DH-PH tandem domains, usually responsible for acceleration of the nucleotide exchange reaction of RHO proteins.

The submitted manuscript describes structural and biochemical characterization of BCR DH and PH domains, respectively. The authors' success in exploiting different biophysical methods and tools for the determination of the DH and PH structures is rather encouraging. Functional analysis have shown that DH does not exhibit any GEF activity neither for non of the analyzed RHO proteins, such as RAC1, CDC42, or RHOA. An assumed sterical clash with binding of RHO proteins by an extended $\alpha 4$ - $\alpha 5$ loop at the edge of the DH domain could unfortunately not be proved neither for the DH nor for the DH-PH proteins. Further analysis have shown that there is no physical interaction between both DH and PH domains. Functional studies on PH domain-membrane interaction have indicated that this domain shows a broad selectivity for phosphoinositides, and is responsible for p210 membrane association, most probably facilitated by F-actin and tyrosine phosphatase STS1.

I believe that the time and care that the authors have invested in this study are remarkable and exemplary. This manuscript is clear and well described. It adds, however, little new insights on understanding the DH domain function for p210. There is a large list of reports with structural and functional investigations on the DH-PH domains of the DBL family. There are meanwhile more than 60 and 190 structures of various DH and PH domains, respectively, available in the database. A lack of GEF activity of the DH domain of BCR and six other DBL family proteins has previously been claimed by Jaiswal et al., JBC (2013). Notably, obtained data are also valid for the BCR proteins itself. Exploring alternative functional role(s) for the DH domain will be very interesting in providing novel mechanistic clues about BCR-ABL function and eventually disease pathogenesis.

Reza Ahmadian

We thank Prof. Ahmadian for his positive evaluation of our manuscript and the useful comment on other DH domains lacking GEF activity. We mention these findings and added a reference to the suggested paper to the Discussion section (page 21).

Reviewer #4 (Remarks to the Author):

Review Sept 15-2017

Structural and functional dissection of the DH and PH domains of oncogenic Bcr-Abl tyrosine kinase.

The authors structurally and functionally characterize the DH and PH domains of Bcr-Abl p210. Differences in phosphorylation and protein interactions between p210 and p190 (a shorter isoform lacking the DH-PH region) have been previously noted, thus the interest in investigating these tandem domains.

Overall the presented work is of high quality and the conclusions drawn are very reasonable. I recommend publication with only minor modifications.

We thank the reviewer for her/his positive evaluation.

The DH domain structure was determined by NMR and by crystallography in complex with a monobody. The PH domain structure was also solved by crystallography in complex with a monobody. SAXS analysis indicated the DH and PH domains in p210 exist in an extended conformation, consistent with their NMR results where chemical shifts of the DH domain were not observed upon addition of the PH domain.

Interaction of RhoA, Rac1 and Cdc42 were not detected with DH and DH-PH protein fragments by pull-down or co-migration through sizing column while Dbs DH-PH served as a positive control. NMR analysis of the Bcr-Abl DH domain revealed a low affinity interface for all three GTPases that mapped to the canonical GTPase binding site. Nucleotide exchange activity of the p210 DH or DH-PH regions was not detected for the three GTPases tested however. The authors declare that Bcr-Abl DH-PH is a pseudo-GEF consistent with a lack of functionally conserved residues in the DH domain sequence compared to other RhoGEFs.

The Bcr-Abl PH domain was shown to bind several phosphoinositide that were reduced by point mutations of basic residues in the expected PIP-binding pocket. Localization of p210 in cells was shown to revert to that of p190 (lacking the DH-PH domain) upon introduction of the PH domain mutations. Co-IP of Bcr-Abl with Sts1 was reduced to the level seen with p190 by these same PH domain mutations in p210.

The authors conclude that the PH domain is primarily responsible for the differential functional properties of p190 compared to p210 based on the PH domain localization role.

Overall, the authors also provide compelling evidence that the purified DH-PH region of Bcr-Abl p210 is a non-functional GEF.

One minor point the authors might comment on in their discussion:

The authors elude to work published by others that RhoGEF activity of Bcr-Abl p210 has been detected and is impaired by a point mutation (S509A). In the absence of detectable GEF activity with purified protein, how do the authors explain the discrepancy with the prior results? With a structure in hand, can an alternate effect of the S509A mutation be predicted (other than loss of GEF activity)?

As we were initially very intrigued by the reported strong effect of the very conservative S509A mutation that was published in the Tala et al 2013 Leukemia paper, we have of course carefully checked the location and a possible function of S509 in the Bcr-Abl DH domain. While S509 is positioned in the CR1 region (see SI Fig. 2F), it is not highly conserved in other DH domain and the residue at this position does not make obvious contacts with bound RhoGTPases in other DH-RhoGTPase complexes. When we expressed and purified the S509A mutant protein, it behaved like the wildtype protein in terms of protein stability and in RhoGTPases binding and

exchange assays. As investigated by NMR, the mutation also does not perturb the structural integrity of the DH domain. We also checked whether S609 could be a phosphorylation site that could modulate RhoGTPase binding/activation: From the very extensive phospho-proteomics datasets that we and many other groups have recorded, S509 was never found phosphorylated, also in line with the lack of a predicted phosphosite consensus sequence. Given all these results, we then revisited the published data on the S509A mutation and found its identification and characterization rather unconvincing and indirect. In addition, strong doubts on the general validity of the findings of the Tala et al 2013 Leukemia paper were expressed by leading scientists in the Bcr-Abl community when I presented our results at different international meetings. In conclusion, I think that the published effects of the S509A mutation are rather dubious. In the revised manuscript, we have now highlighted the position of S509 in SI Fig. 2F and added a cautionary statement to the Discussion section (page 20).

Other items to note:

1) Sequence alignment of DH domains would be helpful to demonstrate that residues important for GTPase interaction, catalysis and selectivity are not conserved in Bcr-Abl p210 compared to other functional Rho GEFs. A sequence alignment of the DH and PH domains would also serve to highlight the loop insertions and various mutations mentioned within the text.

We thank the reviewer for this suggestion. While a sequence alignment of the DH domain highlighting residues important for catalysis and selectivity are shown as SI Figure 2F, we are adding a sequence alignment of the PH domain as well (SI Figure 2E)

2) Are there any other DH domain proteins known without GEF activity? Is this the first pseudo-GEF identified?

Six other Dbl family proteins were proposed to lack GEF activity based on sequence analysis and structural modeling. This is now mentioned in the Discussion section and a respective reference was added (Jaiswal et al. (2013) *J. Biol. Chem.*, 288(6), 4486–4500).